# SLFN11 counteracts the RFWD3-PRIMPOL DNA damage tolerance axis to restrain gapped DNA synthesis in response to replication stress

Kate E. Coleman[1,6], Dong-Woo Shin [2,6], Liana Goehring[1,2,6], Beata Szeitz[1,3], David Fenyö[1,3], Eli Rothenberg [1], John T. Poirier [4,5] ✉ & Tony T. Huang [1] ✉

Schlafen family member 11 (SLFN11) expression sensitizes cells to a spectrum of DNA-damaging chemotherapies. Previous studies have shown that SLFN11 is recruited to stalled replication forks in response to replication stress; however, the role of SLFN11 at stressed replication forks remains unclear. Using single-molecule DNA fiber analysis and super-resolution microscopy to interrogate the dynamics of individual replication forks, we show that SLFN11 acts upon stalled replication forks to suppress efficient fork restart. In the absence of SLFN11 expression, fork restart proceeds through a pathway involving the ubiquitin ligase RFWD3 and the DNA primase-polymerase PRIMPOL to facilitate gapped DNA synthesis, thereby ensuring that cells do not accumulate replication-associated DNA damage. SLFN11 antagonizes this pathway by disrupting recruitment of RFWD3 and PRIMPOL to stalled forks in a manner dependent on a functional ATPase domain and persistent fork localization, but not on tRNA hydrolysis or ssDNA binding. Collectively, our results provide a mechanistic basis for how SLFN11 can counteract DNA damage tolerance by suppressing the RFWD3-PRIMPOL fork restart pathway.

Genome integrity is maintained during DNA replication by cellular processes that have evolved to sense and mitigate replication problems as they arise[1–4]. Replication problems exist on a spectrum from the relatively benign, such as temporary insufficiency in nucleotide availability, to the severe, such as DNA breaks that produce free ends[5,6]. To restore replication fork progression, it may be necessary to stabilize stalled forks, metabolize reversed forks, repair a variety of types of DNA lesions, and orchestrate the recruitment of factors required to restart replication[7–9].

Replication stress has been defined as the transient slowing or stalling of replication forks[10]. Stalled replication forks must be stabilized to prevent collapse into toxic double-strand breaks (DSBs), which requires timely removal of any replication impediment, followed by fork restart and subsequently fork elongation[11,12]. Replication stress is sensed by the accumulation of RPA on single-stranded DNA, which activates the ATR-CHK1 replication stress response signaling pathway to coordinate responses local to stalled forks and global to dormant origins firing and cell cycle checkpoint activation[13]. Two landmark studies identified high expression of Schlafen family member 11 (*SLFN11*) as a predictive marker of sensitivity to pharmacologic perturbations that induce replication stress[14,15]. Subsequently, forward

[1]Department of Biochemistry & Molecular Pharmacology, New York University Grossman School of Medicine, New York, NY, USA. [2]Vilcek Institute of Graduate Biomedical Sciences, NYU Grossman School of Medicine, New York, USA. [3]Institute for Systems Genetics, New York University Grossman School of Medicine, New York, NY, USA. [4]Department of Medicine, Laura and Isaac Perlmutter Cancer Center, New York University Grossman School of Medicine, New York, NY, USA. [5]Laura and Isaac Perlmutter Cancer Center, New York University Langone Health, New York, NY, USA. [6]These authors contributed equally: Kate E. Coleman, Dong-Woo Shin, Liana Goehring. ✉e-mail: John.Poirier@nyulangone.org; Tony.huang@nyulangone.org

genetic screens identified SLFN11 as a functional sensitizer to exogenous and endogenous sources of replication stress[16,17].

SLFN11 is a group III schlafen that encodes an N-terminal endonuclease domain, a linker domain, and a putative C-terminal helicase domain and self-assembles as a homodimer[18,19]. The endonuclease domain cleaves type II tRNAs, which was first shown to restrict replication of several viruses[20–23]. In the context of replication stress, depletion of type II tRNAs can disrupt translation of proteins with maladaptive codon usage[24], induce proteotoxic stress[25] and cause ribosome stalling[26], leading to apoptosis.

The C-terminal region of SLFN11 encodes a putative helicase with unknown substrate and unclear ATPase activity. The C-terminal region of SLFN11 has also been shown to bind to ssDNA[27,28], and to interact with the 70 kDa subunit of RPA to displace the complex from resected DNA ends after a double-strand break[29]. Recombinant SLFN11 can neither bind nor hydrolyze ATP[27]; however, the deletion of the C-terminal domain or the introduction of inactivating mutations into the Walker A and B motifs abolish SLFN11 activity in complementation assays[29,30]. Under conditions of replication stress, ATPase activity was required to promote degradation of the replication licensing factor CDT1[31], enhance stalled replication fork degradation[32], and irreversibly block DNA replication[30]. Despite these recent studies on SLFN11, it is still unclear if and how SLFN11 regulates replication stress recovery in cells.

In this work, we sought to determine how SLFN11 regulates replication fork dynamics in response to replication stress. Using single-molecule methods, we show that SLFN11 antagonizes a fork recovery pathway downstream of ATR that involves the ubiquitin ligase RFWD3 and the human DNA primase-polymerase (PRIMPOL). In the absence of SLFN11, this pathway promotes gapped DNA synthesis, thereby preventing stalled replication forks from undergoing replication-associated DNA damage. RFWD3 has a documented role in promoting fork restart[33–40], while recent studies have implicated PRIMPOL as a generalized translesion synthesis (TLS) polymerase with the unique function of initiating de novo DNA synthesis[41]. Using single-molecule localization microscopy (SMLM-STORM) imaging techniques, we discovered that persistent localization of SLFN11 to restarted forks suppresses RFWD3 and PRIMPOL recruitment, and that PRIMPOL localization to replication forks was largely dependent on RFWD3. These findings demonstrate that SLFN11 compromises replication fork restart by inhibiting a previously under-appreciated RFWD3-PRIMPOL DNA damage tolerance axis.

## Results

### SLFN11 suppresses ubiquitination of the RFWD3 substrate RPA leading to impaired fork restart

HAP1 is a near-haploid human cell line derived from the KBM-7 chronic myelogenous leukemia (CML) cell line that expresses SLFN11, and acquires resistance to perturbations that induce replication stress upon SLFN11 loss[17,32,42]. We used hit-and-run CRISPR gene disruption[43,44] in multiple constitutive coding exons of *SLFN11*, followed by single-cell cloning to generate clones with frameshift indels that result in complete loss of SLFN11 protein expression (Fig. 1A, Supplementary Fig. 1A). To determine the role of SLFN11 in regulating replication fork dynamics, we interrogated individual replication forks using the well-established single-molecule DNA fiber assay[45]. To test whether the loss of SLFN11 expression affected replication fork speed in the absence of perturbation, we sequentially pulse-labeled HAP1 parental and *SLFN11* KO clones with IdU and CldU nucleoside analogs to determine DNA tract extension over a 20 min period (Fig. 1A). In this assay, the initial IdU pulse marks active replication forks, while the subsequent CldU pulse marks nucleotide incorporation over a defined time frame. Only tracts with IdU (green) labeling are counted, and only the length of the CldU (red) second label is measured, ensuring that newly fired replication origins are not considered. We observed no difference in tract length as a surrogate for replication fork speed

between parental and *SLFN11* KO cell lines. We also observed no detectable changes in the total DNA synthesis levels of parental *versus SLFN11* KO cells, as measured in individual cells by EdU incorporation followed by flow cytometry (Supplementary Fig. 1B). These results are consistent with a lack of any significant DNA replication defect in the unperturbed, SLFN11-expressing or *SLFN11* KO HAP1 cell line.

Next, to determine whether SLFN11 expression plays a functional role in recovering from replication stress perturbations, we pulsed HAP1 parental and *SLFN11* KO clones with IdU to mark active replication forks, after which replication was transiently arrested using high-dose HU (2 mM) to deplete the deoxyribonucleoside triphosphate (dNTP) pool. Subsequently, HU was washed out and replaced with CldU-containing media to label forks that were able to restart DNA synthesis. We observed a dramatic increase in CldU tract length in *SLFN11* KO lines (Supplementary Fig. 2A). A more accurate measurement of tract length changes is to display the results as the ratio of CldU/IdU tract length, which is proportional to changes in replication fork speed before and after restart. As expected, we also observed a striking increase in the CldU/IdU ratio in *SLFN11* KO lines, in comparison to parental cells, consistent with unrestrained DNA synthesis (Fig. 1B). Depletion of SLFN11 using two independent siRNAs in HAP1 cells recapitulated the effects observed in *SLFN11* KO lines, ruling out potential off-target effects of CRISPR-Cas9 gene disruption or clonal artifacts (Supplementary Fig. 2B). Although the majority of HU-induced stalled replication forks were competent for fork restart irrespective of SLFN11 expression, significantly fewer forks remained stalled in the *SLFN11* KO clones (Supplementary Fig. 2C), suggesting that fork restart is less efficient in cells that express SLFN11.

SLFN11 has been reported to disrupt sustained ATR checkpoint activation under chronic replication stress[29]. Given findings indicating that SLFN11-negative cells can be sensitized to a variety of chemical perturbations by ATR inhibition[30,31,46,47], we tested whether ATR signaling in cells is affected by SLFN11 expression at early time points following induction of replication stress. HAP1 parental or *SLFN11* KO lines were treated with 2 mM HU for 4 h to induce replication stress. Western blot analysis of cell lysates revealed robust phosphorylation of ATR substrates CHK1 at Ser345, as well as RPA32 at Ser33 and S4/S8 sites, irrespective of SLFN11 expression, indicating expected activation of ATR in response to replication stress. While we observed no consistent differences in the phosphorylation of ATR substrates, we did, however, observe a loss of RPA32 ubiquitination (Ub-RPA32) in the HAP1 parental, SLFN11-expressing cells upon induction of replication stress (Fig. 1C).

We next sought to determine whether reconstitution of SLFN11 expression back into the *SLFN11* KO cells would restore the fork restart defect and reduce Ub-RPA32 observed in the parental cell line. Toward that end, we generated a doxycycline (dox)-inducible cDNA system to rescue SLFN11 protein expression levels in HAP1 *SLFN11* KO cell lines. Partial rescue of SLFN11 expression was sufficient to reduce Ub-RPA32 levels in HU-treated cells (Fig. 1D). Consistent with these findings, the reconstitution of SLFN11 expression was sufficient to restore the parental fork restart defect in DNA fiber assays (Fig. 1E, Supplementary Fig. 2D). SLFN11-dependent suppression of Ub-RPA32 was independently confirmed in HEK293T cells, which lack endogenous SLFN11 expression, using a dox-inducible SLFN11 system. Following transfection with FLAG-ubiquitin and treatment with HU, Ub-RPA32 levels were markedly reduced in SLFN11-expressing cells compared to parental cells (Fig. 1F). These observations demonstrate that SLFN11 expression reduces RPA32 ubiquitination and produces a defect in fork restart in response to replication stress without disrupting phosphorylation of canonical ATR substrates.

Ubiquitin signaling regulates the activity of several proteins at replication forks and orchestrates the temporal recruitment of different proteins as recovery from replication stress proceeds (reviewed in Mirsanaye et al[48]). RPA is known to be ubiquitinated by a single E3

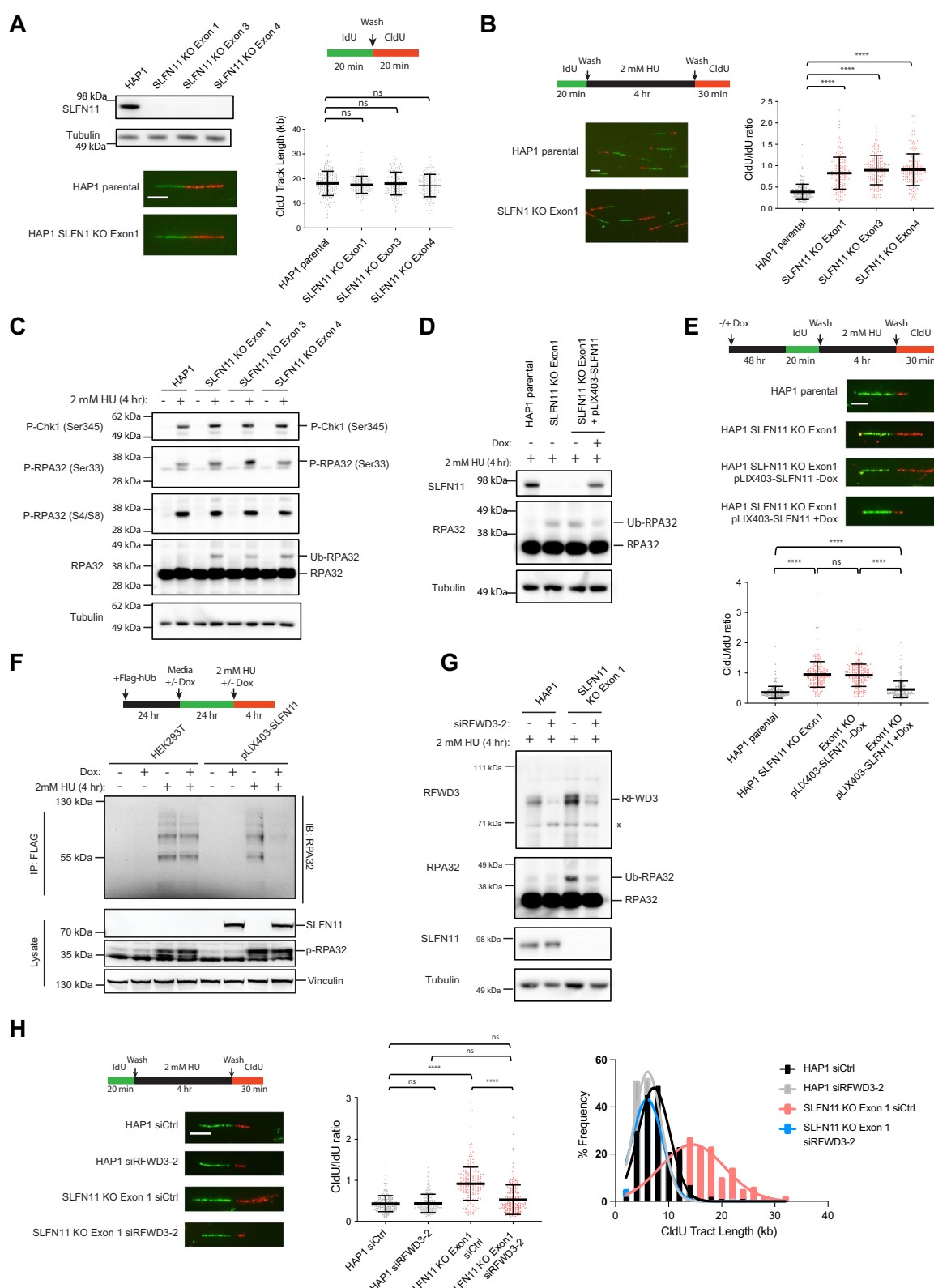

ligase: RING finger and WD repeat domain 3 (RFWD3/FANCW)[33–35,49–51]. Familial missense mutations in RFWD3 abolish RPA ubiquitination and cause Fanconi anemia (FA), which is characterized by defects in interstrand crosslink repair[52,53]. RFWD3 is required for ubiquitination of all subunits of RPA, and reciprocal mutations that disrupt the interaction between the WD repeat of RFWD3 and the winged-helix domain of RPA32 lead to loss of RPA ubiquitination, defective

replication fork restart, and recapitulate the FA phenotype in cell lines and in *Rfwd3* knockout mice[51].

To test whether unrestrained DNA replication after fork-stalling events is functionally linked to RFWD3 in *SLFN11* KO cells, we employed both siRNA knockdown and CRISPR-Cas9 strategies to disrupt RFWD3 protein expression in HAP1 cells. Consistent with its essential role in RPA ubiquitination, siRNA knockdown of RFWD3 in

**Fig. 1 | SLFN11 restrains replication fork restart and DNA synthesis in response to replication stress. A** Western blot analysis of parental HAP1 cells and individual HAP1 SLFN11 KO clones with the indicated antibodies; for this and all subsequent experiments, tubulin serves as a loading control. Schematic for measuring replication fork elongation tract length (fork speed) in unperturbed cells by DNA fiber analysis. A representative image of elongating tracts from HAP1 parental and SLFN11 KO cells; Scale bar = 5 μm. Swarm plot of CldU tract length measurements from elongating forks in the indicated cells from three biological replicates ($n = 200$ elongating forks) with mean and ± standard deviation (SD) indicated. p-values were calculated using the Mann-Whitney rank-sum t-test (ns = no significance, two-tailed). **B** Schematic for measuring fork restart and DNA synthesis (fork length) following recovery after treatment with 2 mM HU for 4 h by DNA fiber analysis. Representative restarted tracts of HAP1 parental and SLFN11 KO cells; Scale bar = 5 μm. Swarm plot of the ratio of CldU/IdU tract lengths from the fork restart assay. Data are plotted for three biological replicates ($n = 180$ restarted forks) with mean and ± SD indicated. *p*-values were calculated using the Mann-Whitney rank-sum *t*-test (****p < 0.0001, two-tailed). **C** Western blot analysis of parental HAP1 and SLFN11 KO cells with 2 mM HU for 4 h with the indicated antibodies. **D** Parental HAP1, SLFN11 KO, and SLFN11 KO cells reconstituted with doxycycline (Dox)-inducible SLFN11 expression (pLIX403-SLFN11) for 48 h were treated with 2 mM HU for 4 h as indicated were analyzed by Western blot. **E** Schematic of DNA fiber analysis for measuring fork restart and DNA synthesis in Dox-induced conditions following HU recovery. Representative images of restarted forks from the indicated cell lines treated; Scale bar = 5 μm. Swarm plot showing the ratio of CldU/IdU tract lengths from the fork restart assay. Data are plotted for three biological replicates ($n = 180$ restarted forks) with mean and -/+ SD indicated. *p*-values were calculated using the Mann-Whitney rank-sum *t*-test (ns = no significance, ****p < 0.0001, two-tailed). **F** Schematic of the experimental workflow used to assess Ub-RPA32 levels in HEK293T cells following HU treatment, comparing cells with Dox-inducible SLFN11 expression. Immunoblot of RPA32 following FLAG-Ub IP to enrich for ubiquitinated proteins. **G** Parental HAP1 and SLFN11 KO cells after RFWD3 siRNA knockdown for 72 h were treated with 2 mM HU for 4 h. Cell lysates were probed with the indicated antibodies for Western blot analysis. Asterisks signify non-specific bands. **H** Representative images of restarted forks from HAP1 and SLFN11 KO cells in -/+ RFWD3 knockdown are shown. Swarm plot showing the ratio of CldU/IdU tract lengths from the fork restart assay (see schematic). Data are plotted for three biological replicates ($n = 180$ restarted forks) with mean and -/+ SD indicated. *p*-values are calculated using the Mann-Whitney rank-sum *t*-test (ns = no significance, ****p < 0.0001, two-tailed). Histogram of all the CldU tract lengths of restarted elongating forks from cells treated after -/+ RFWD3 siRNA knockdown (below). Scale bar = 5 μm.

HAP1 *SLFN11* KO cells reduced Ub-RPA32 levels (Fig. 1G). siRNA knockdown of RFWD3 alone in HAP1 parental cells that express SLFN11 had little to no effect on IdU tract lengths or replication fork restart as quantified by CldU tract lengths or as CldU/IdU ratio (Supplementary Fig. 2F, Fig. 1H). However, in *SLFN11* KO cells, loss of RFWD3 dramatically reduced fork restart to levels similar to those observed in HAP1 parental cells after HU treatment (Fig. 1H, Supplementary Fig. 2F). Similarly, *RFWD3* KO clones generated from *SLFN11* KO cell lines (Supplementary Fig. 2E) phenocopies HAP1 parental, SLFN11-expressing cells (Supplementary Fig. 2G). Together, these results show that RFWD3 is required for replication fork restart in cells lacking SLFN11 expression.

## Restarted forks contain ssDNA gaps in response to replication stress in the absence of SLFN11

The resolution of a standard DNA fiber assay does not distinguish between fully contiguous labeling and discontinuous labeling interspersed with short ssDNA gaps. To determine whether the restarted replication forks were continuous or contained ssDNA gaps after HU treatment, we treated the labeled, nascent DNA with the ssDNA-specific S1 nuclease. We performed the HU fork restart assay in parental HAP1 cells or *SLFN11* KO cells with the addition of S1 nuclease treatment at the termination of the experiment. S1 nuclease treatment reduced the CldU/IdU ratio (Fig. 2A) and overall CldU tract length (Supplementary Fig. 3A) in *SLFN11* KO cell lines, consistent with the presence of ssDNA gaps. The specificity of the S1 nuclease treatment is internally controlled by the IdU tract, which shows no difference in tract length under any condition (Supplementary Fig. 3B). These data demonstrate that the extended DNA tracts, which represent the unrestrained DNA synthesis after replication stress in *SLFN11* KO cells, are interspersed with ssDNA gaps.

## Unrestrained replication fork restart requires the primase-polymerase PRIMPOL

The genesis of ssDNA gaps could be explained by a variety of potential mechanisms, including lagging-strand defects, repriming of leading-strand synthesis, or failure of gap suppression mechanisms (reviewed in Cong & Cantor[54]). Eukaryotic translesion polymerases have evolved to bypass specific types of DNA lesions, enabling fork restart (reviewed in Waters et al[55].). It has recently been appreciated that the human DNA primase-polymerase (PRIMPOL) can bypass DNA lesions entirely by de novo primer synthesis, a feature of no other TLS polymerase (reviewed in Guilliam & Doherty[41]).

We hypothesized that the role of PRIMPOL as a generalized DNA lesion bypass and replication-coupled mechanism could be a likely explanation for HU-induced ssDNA gaps during unrestrained DNA synthesis when SLFN11 is not expressed. To test this hypothesis, we knocked down PRIMPOL in HAP1 parental and *SLFN11* KO cell lines, then treated cells with 2 mM HU for 4 h. Western blot analysis confirmed that PRIMPOL was significantly depleted, which had no effect on Ub-RPA32 levels (Fig. 2B). Functionally, PRIMPOL knockdown in *SLFN11* KO cells phenocopies RFWD3 knockdown in *SLFN11* KO cells with respect to CldU/IdU ratio and CldU tract length in the context of replication fork restart (Fig. 2C). However, PRIMPOL knockdown further exacerbates the replication fork restart defect as measured by a significant increase in the fraction of stalled forks (Supplementary Fig. 3C). These findings were recapitulated using an alternative siRNA sequence (Supplementary Fig. 3D), as well as in different cancer cell lines, including the lung cancer cell line SW1271, and the prostate adenocarcinoma cell line DU145 (Supplementary Fig. 4A, B). These cancer cell lines of different lineages natively express SLFN11 at levels similar to HAP1 cells. Supporting a functional role for this pathway, cell viability assays showed that disruption of either PRIMPOL or RFWD3 in *SLFN11* KO cells significantly reduced HU resistance, restoring the $EC_{50}$ toward that of the parental cell line (Fig. 2D). Taken together, these data indicate that ssDNA gaps that are present in restarted forks in various SLFN11-deficient cell lines are likely dependent on PRIMPOL engagement at restarted forks.

## PRIMPOL loss leads to elevated replication-associated DNA damage

Pan-cancer drug sensitivity profiles suggest that the role of SLFN11 is not lineage specific and instead exerts effects that are common to multiple different biological contexts[14,15]. The osteosarcoma cell line U2OS and the retinal pigment epithelial cell line hTERT RPE-1 are two of the most thoroughly characterized cell lines in the DNA damage/repair field. They represent a cancer cell line with loss of genome integrity and an hTERT immortalized cell line with normal karyotype[56–58]. Both cell lines express PRIMPOL, which can be efficiently knocked down by siRNA, but lack endogenous SLFN11 expression (Supplementary Fig. 4C). PRIMPOL knockdown in either cell line was sufficient to eliminate extended CldU tracts, which are indicative of unrestrained DNA synthesis during replication stress recovery (Supplementary Fig. 4D). Additionally, PRIMPOL knockdown in RPE cells led to more stalled forks, similar to the fork restart defect observed in HAP1 cells (Supplementary Fig. 4E). Importantly, ectopic expression of SLFN11 using doxycycline (dox)-inducible

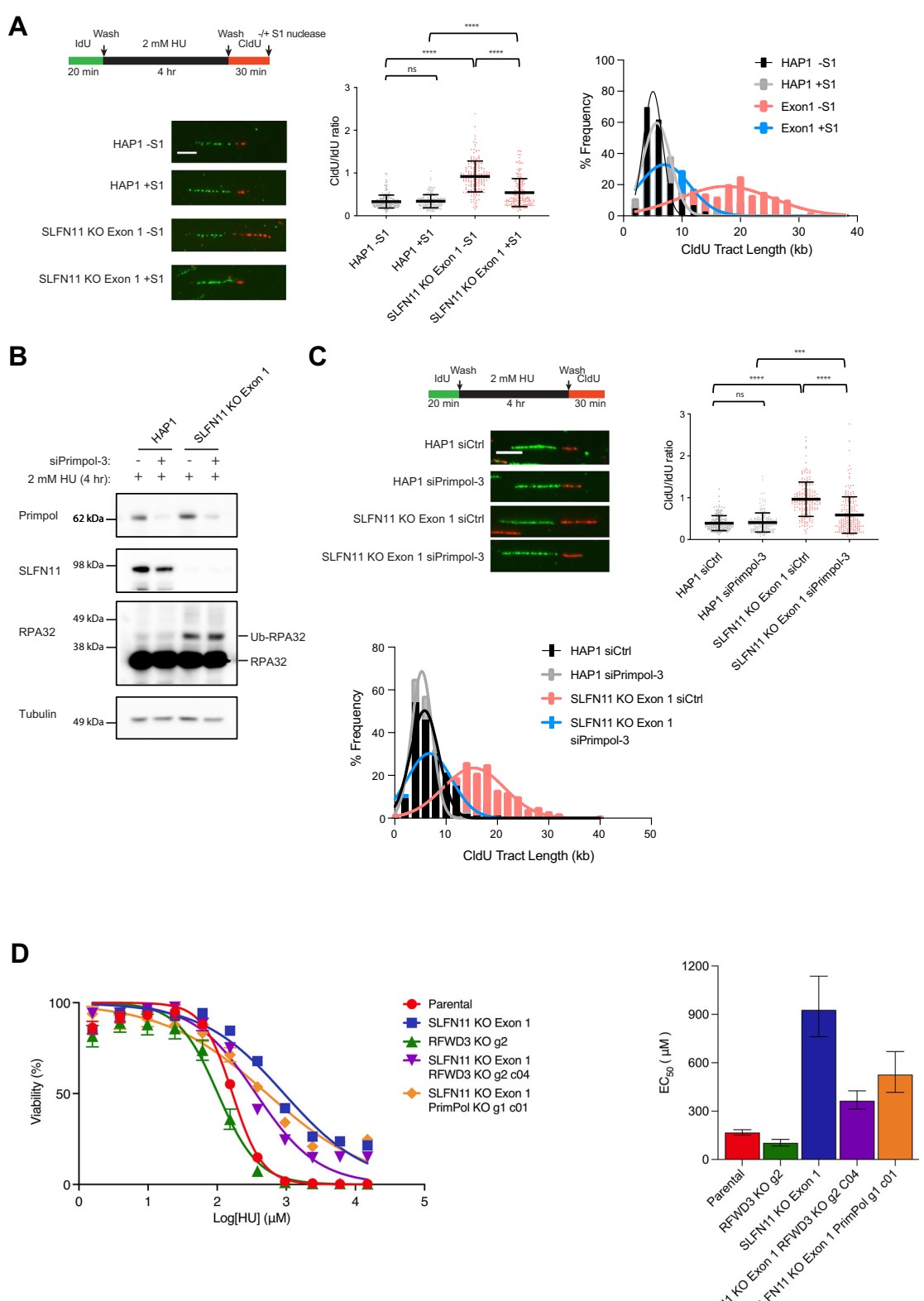

expression system in RPE-1 cells (RPEs) (Fig. 3A) is sufficient to cause the replication fork defect that was observed in cell lines that natively express SLFN11 (Supplementary Fig. 4A, B).

To gain deeper, mechanistic insight into how SLFN11 restrains DNA synthesis, we used a multi-color Single-Molecule Localization Microscopy (SMLM) approach to directly visualize SLFN11-dependent nascent DNA, RFWD3, and PRIMPOL fluorescently-labeled molecules

in the setting of fork restart. The increased sensitivity and spatial resolution of SMLM provide nanometer-scale features of protein localization at replication forks, which cannot be as accurately resolved via diffraction-limited fluorescence microscopy[59]. Replication forks in untreated RPE cells or those treated with dox to induce SLFN11 expression were stalled by HU, washed with PBS, and released into fresh media containing EdU (Fig. 3B). Nuclei were prepped for SMLM

**Fig. 2 | PRIMPOL-mediated ssDNA gaps underlie accelerated fork progression following fork restart in the absence of SLFN11. A** Parental HAP1 and SLFN11 KO cells were subjected to the fork restart assay, and the isolated DNA fibers were treated with or without S1 nuclease to digest ssDNA to reveal gapped DNA synthesis (see schematics and representative images). Swarm plot showing the ratio of CldU/IdU tract lengths from the fork restart assay with HU, -/+ subsequent S1 nuclease digestion. Data are plotted from three biological replicates ($n = 180$ restarted forks) with mean ± SD indicated. $p$-values were calculated using the Mann-Whitney rank-sum $t$-test (ns = no significance, ****$p < 0.0001$, two-tailed). Scale bar = 5 μm. **B** Parental HAP1 and SLFN11 KO cells were treated after -/+ Primpol siRNA knockdown for 72 h and incubated with 2 mM HU for 4 h. Cell lysates were probed with the indicated antibodies for Western blot analysis. **C** Scatter plot showing the ratio of CldU/IdU tract lengths from the fork restart assay (see schematic). Data are plotted from three biological replicates ($n = 180$ restarted forks) with mean ± SD indicated. $p$-values were calculated using the Mann-Whitney rank-sum $t$-test (ns = no significance, ***$p < 0.001$, ****$p < 0.0001$, two-tailed). Representative images of restarted forks from parental HAP1 and SLFN11 KO cells in -/+ Primpol siRNA knockdown are shown. Scale ba = 5 μm. Histogram of CldU tract lengths of restarted elongating forks from HAP1 parental or SLFN11 KO cells after -/+ Primpol siRNA knockdown. **D** Dose-response curves showing cell viability for HAP1 parental, SLFN11 KO, RFWD3 KO, SLFN11/RFWD3 DKO, and SLFN11/PRIMPOL DKO cells following 48 h treatment with HU (1.57 μM to 15 mM). Data points represent the mean of five biological replicates; error bars indicate -/+ SEM (left). $EC_{50}$ values calculated by nonlinear regression using the log[HU] $vs$ normalized response (variable slope) in GraphPad Prism. Error bars represent the 95% confidence interval of the calculated $EC_{50}$ (right).

---

analysis, including co-staining with PCNA to mark S-phase nuclei and actively replicating regions of interest. Quantification of the average EdU density in PCNA(+) nuclear regions of interest revealed results complementary to our DNA fiber results, wherein cells that do not express SLFN11 have unrestrained EdU incorporation following restart compared to SLFN11-expressing RPEs (Fig. 3B). Consistent with a role for RFWD3 and PRIMPOL in mediated fork restart in SLFN11-negative cells, knockdown of either RFWD3 or PRIMPOL similarly revealed defective DNA synthesis following release from HU in SLFN11-negative RPEs (Fig. 3C, D).

To determine whether restrained DNA synthesis in SLFN11-expressing cells contributes to elevated replication-associated DNA damage, we quantified the average gH2AX (pS139-H2AX) density at nascent DNA marked by EdU incorporation in PCNA(+) nuclei by SMLM (Fig. 3E–G). SMLM analysis of replication fork restart dynamics showed that knockdown of PRIMPOL does not impact DNA synthesis (EdU density at PCNA sites) in RPE cells harboring dox-induced SLFN11 expression, suggesting that PRIMPOL-mediated gapped DNA synthesis functions primarily in the absence of SLFN11 (Fig. 3F). Next, we found that increased DNA damage, as marked by gH2AX signal, was associated with forks that cannot fully restart due to either pathway suppression by SLFN11 or when PRIMPOL was knocked down in cells lacking SLFN11 expression (Fig. 3G). Therefore, cells that can undergo fork restart and unrestrained DNA synthesis in the absence of SLFN11 expression displayed less replication-associated DNA damage due to the presence of PRIMPOL. Collectively, these results show that PRIMPOL mitigates replication-associated DNA damage, in part, through the promotion of fork restart and unrestrained DNA synthesis in response to stalled replication forks.

### Persistent SLFN11 localization to RPA inhibits replication fork restart

To determine whether SLFN11 directly antagonizes replication fork restart, we analyzed by SMLM the ability of SLFN11 to dynamically bind replication forks in untreated, HU, and restart conditions (Fig. 4A). Previous work from our group and others showed that SMLM nuclear preparation, which involves CSK extraction prior to fixation and antibody labeling, minimizes detection of nuclear soluble and cytoplasmic species[60]. Therefore, SMLM detection of SLFN11 is primarily chromatin-bound. RPEs with and without dox treatment were subjected to single-molecule detection using antibodies against RPA and SLFN11 (Fig. 4A). Compared to the noise/unspecific signal in untreated RPEs, Dox-treated RPEs has higher levels of chromatin-bound SLFN11 (Fig. 4B). SLFN11 binding to chromatin is increased upon treatment with HU, and even further elevated following fork restart compared to HU-treatment (Fig. 4B). The average density of SLFN11 at RPA is similarly increased upon HU treatment compared to no HU in dox-treated RPEs, and again further elevated upon restart conditions compared to HU treatment (Fig. 4C). This suggests that SLFN11 associates with RPA at stalled forks in response to replication stress.

Previous work has characterized putative functional domains of SLFN11, including RNA hydrolysis/endonuclease, ATPase/Walker B motif, and single-stranded DNA-RPA binding activities[26,27,29,30,61,62] (Fig. 4D). RPEs with dox-inducible expression of SLFN11-mutants were generated and analyzed by SMLM for fork restart efficiency. Interestingly, expression of the SLFN11 mutants carrying point mutations that disrupt either tRNA endonuclease activity (E209A/E214A), or ssDNA binding (K652D) phenocopied dox-induced expression of WT-SLFN11 in blocking replication restart, suggesting that activity is required for SLFN11 to inhibit fork restart (Fig. 4E, F). Consistent with our findings that tRNA hydrolysis was not required for the fork restart defect, we observed no measureable changes to endogenous RFWD3 protein levels with or without SLFN11 expression (Supplementary Fig. 5A). In contrast to the other SLFN11 mutants, expression of mutant-SLFN11 with point mutations in its ATPase domain (K605M/E669Q) was unable to block fork restart, as measured by SMLM (Fig. 4G). Intriguingly, the chromatin binding of the SLFN11 ATPase mutant upon HU treatment is similar to WT-SLFN11 except when following fork restart, when the SLFN11 ATPase mutant fails to associate with restarted forks (Supplementary Fig. 6A). Further, in contrast to WT-SLFN11, chromatin localization of the SLFN11 ATPase mutant does not correlate with RPA binding (Supplementary Fig. 6B). This suggests that persistent and enhanced chromatin localization of WT-SLFN11 is mediated, in part, through its ATPase domain via RPA binding at the fork in response to replication stress in order to disrupt replication fork restart.

### SLFN11 antagonizes RFWD3-dependent recruitment of PRIMPOL to restarted forks

We next hypothesized whether persistent localization of SLFN11 to restarted replication forks could disrupt the recruitment of both RFWD3 and PRIMPOL. To address this, we used SMLM to detect the recruitment of FLAG-tagged wildtype (WT-FLAG-RFWD3) and RING domain mutated (C315A) RFWD3 (mutant-FLAG-RFWD3) to PCNA in no dox- or dox-treated SLFN11-expressing RPE cells (Fig. 5A). Following release from HU, SLFN11-expressing RPE cells revealed defective recruitment of both WT- and mutant-FLAG-tagged RFWD3 to restarted forks when compared to no dox-treated cells that lack SLFN11 expression (Fig. 5B). In contrast, expression of the SLFN11 ATPase mutant is unable to inhibit either WT- or mutant-FLAG-RFWD3 recruitment to PCNA during restart (Supplementary Fig. 6C). To further explore how SLFN11 alters the recruitment of RFWD3 to chromatin and replication forks, we performed anti-FLAG antibody immunoprecipitation followed by mass spectrometry (IP/MS) using the mutant-FLAG-RFWD3, which resists auto-ubiquitination and degradation (Fig. 5C). Mutant-FLAG-RFWD3 (bait) pulldown was efficient in all samples. However, after HU treatment, SLFN11-expressing HAP1 parental cells showed markedly reduced association with DNA-binding and DNA repair proteins including previously nominated RFWD3 interactors such as FANCD2, DNAJC13, YY1, GTF3C4, MED12, PIAS1, and TONSL compared to SLFN11 KO cells[63] (Fig. 5C, Supplementary Fig. 7A).

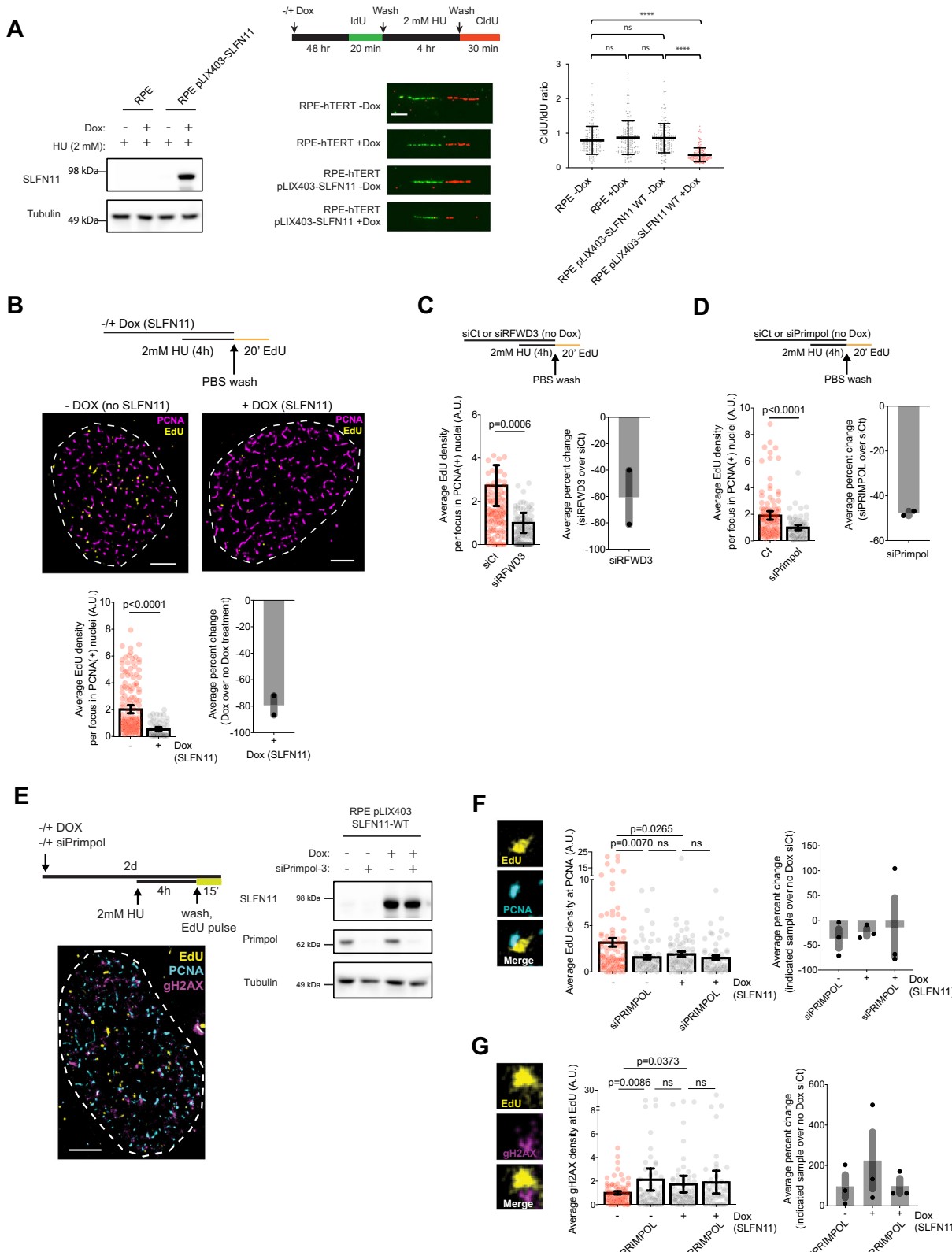

These findings show that SLFN11 may play a broader role in restricting the access of the E3 ligase RFWD3 to chromatin/replication forks, thereby limiting RFWD3's ability to modulate DNA repair and fork recovery processes–consistent with reduced ubiquitination of its substrate RPA32.

To determine how SLFN11 and RFWD3 expression impact PRIM-POL recruitment to forks during fork restart, SMLM analysis was performed using extracted nuclei immunostained for PRIMPOL and PCNA following HU restart and EdU-pulse labeling during release into fresh media (Fig. 5D). Dox-induced expression of SLFN11 in RPEs decreased PRIMPOL recruitment to forks (Fig. 5D). In agreement with the role of PRIMPOL in mediating gapped DNA synthesis, the average EdU density at localized PRIMPOL was decreased with SLFN11 expression (Fig. 5E), suggesting that PRIMPOL could be responsible for

**Fig. 3 | PRIMPOL prevents replication stress-associated DNA damage in the absence of SLFN11. A** (Left) parental RPE-1 and RPE-1 pLIX403-SLFN11 cells (-/+ Dox for 48 h) were treated with 2 mM HU for 4 h. Cell lysates were analyzed by Western blot with the indicated antibodies. (Right) Schematic for fork restart assay to measure fork speed by DNA fiber analysis. Representative images of restarted forks from treated cells; scale bar = 5 μm. Swarm plot shows the ratio of CldU/IdU tract lengths following release from HU is based on three biological replicates (n = 180 restarted forks) with mean and -/+ SD indicated. p-values were calculated using the Mann-Whitney rank-sum t-test (ns = no significance, ****p < 0.0001, two-tailed). **B** (Left) Representative SMLM image shows -/+ Dox treatment (WT-SLFN11 expression) of RPEs stained with PCNA (magenta) and EdU (yellow) following fork-stalling with 2 mM HU, and release into fresh media with EdU following washout (see "Methods"). Scale bar = 2 μm. (Bottom, left) Scatterplot quantification measuring the average EdU density per focus in PCNA+ nuclei in -/+ SLFN11-expressing RPEs. p-values of technical replicates calculated using an unpaired two-tailed t-test from two biological replicates (Unt: N = 155, DOX: N = 55). Error bars = mean, SEM. (Bottom, right) Bar graph showing percent change between the average of two biological replicates, each normalized to the lowest mean condition relative to untreated RPEs. **C** RPEs were treated with -/+ RFWD3 siRNA knockdown (no Dox). HU-induced fork-stalling, washout, and release into fresh media with EdU was performed prior to SMLM analysis with PCNA and EdU. (Left) Scatterplot quantification measuring the average EdU density per focus in PCNA+ nuclei in -/+ RFWD3 knockdown RPEs. p-values of technical replicates calculated using an unpaired two-tailed t-test from two biological replicates (siCt: N = 131, siRFWD3: N = 173). Error bars = mean, SEM. (Right) Bar graph showing the percent change between the average of two biological replicates, each normalized to the lowest mean condition relative to siCt. **D** -/+ PRIMPOL siRNA knockdown (no Dox) RPEs were treated and analyzed as in (**C**). (Left) Scatterplot quantification measuring the average EdU density per focus in PCNA+ nuclei in -/+ PRIMPOL siRNA knockdown RPEs. p-values of technical replicates calculated using an unpaired two-tailed t-test from two biological replicates (siCt: N = 112, siPRIMPOL: N = 72). Error bars = mean, SEM. (Right) Bar graph showing the percent change between the average of two biological replicates, each normalized to the lowest mean condition relative to siCt. **E** (Left) Representative SMLM image of RPE-1 pLIX403-SLFN11 cells after -/+ PRIMPOL siRNA and -/+ Dox as indicated. Cells were treated with 2 mM HU for 4 h to stall forks and released into fresh media with EdU to allow recovery (see schematic). Cells were stained with PCNA (cyan) and gH2AX (magenta) for SMLM. Scale bar = 2 μm. (Right) RPE-1 pLIX403-SLFN11 cells with -/+ PRIMPOL siRNA and -/+ Dox as indicated to confirm SLFN11 protein expression and PRIMPOL knockdown efficiency in RPEs by Western blot. **F** (Left) Scatterplot quantification measuring the average EdU density at PCNA using pair correlation analysis (see "Methods"). p-values of technical replicates calculated using an unpaired two-tailed t-test from two biological replicates (Unt: N = 96, siPRIMPOL: N = 66, DOX: N = 78, DOX+siPRIMPOL: N = 58). Error bars = mean, SEM. (Right) Bar graph showing the percent change between the average of two biological replicates, each normalized to the lowest mean condition relative to siCt. **G** SMLM analysis of average gH2AX density at EdU. p-values of technical replicates calculated using an unpaired two-tailed t-test from two biological replicates (Unt: N = 96, siPRIMPOL: N = 66, DOX: N = 78, DOX+siPRIMPOL: N = 58). Error bars = mean, SEM. (Right) Bar graph showing the percent change between the average of two biological replicates, each normalized to the lowest mean condition relative to siCt.

leading strand re-priming at restarted forks; this would likely attenuate replication-associated DNA damage (Fig. 3G). Further, SMLM analysis revealed that PRIMPOL recruitment to replication forks is compromised by the knockdown of RFWD3 (Fig. 5F, G), suggesting that RFWD3 functions upstream of PRIMPOL for fork restart. This is consistent with our findings that knockdown of PRIMPOL did not impact RFWD3-mediated RPA ubiquitination (Fig. 2B). Interestingly, knockdown of RFWD3 in SLFN11-expressing RPE cells did not further reduce PRIMPOL localization at forks (Fig. 5G). This suggests that PRIMPOL's function during fork restart can be suppressed by RFWD3 depletion, but only when SLFN11 is unavailable. Taken together, these data support a model in which PRIMPOL functions downstream of RFWD3 (see model, Fig. 6A). As a side note, SLFN11 protein levels were noticeably decreased after RFWD3 knockdown. We only observed this effect in RPE-1 cells, but not in other cell lines; this does not change the conclusion of this study.

## Discussion

In this study, we sought to understand how SLFN11 expression affects the local stalled replication fork response to replication stress. We observed reduced RFWD3 and PRIMPOL localization at elongating forks in response to replication stress in cells where SLFN11 is expressed. Concordantly, we determined that SLFN11-expressing cells are significantly impaired in PRIMPOL-mediated replication fork restart, leading to an increase in replication fork-associated DNA damage and genomic instability. SLFN11 expression could be phenocopied, with respect to the lack of proper replication fork restart/elongation, by loss of either RFWD3 or PRIMPOL. These findings provide strong evidence that SLFN11 inhibits replication fork restart and highlight a previously unrecognized regulatory axis between RFWD3 and PRIMPOL in the fork recovery pathway. We propose that RFWD3 and PRIMPOL form a DNA damage tolerance pathway axis to counteract replication lesions and the toxicity inflicted by chemotherapeutic drugs, that is antagonized by SLFN11 expression.

RFWD3, as an FA gene, is known to play a key role in replication fork restart mediated by ATR and interstrand crosslink repair[33–37,39,49–53,64]. These functions are tied to the E3 ubiquitin ligase activity of RFWD3 and ubiquitination of its substrates, including members of the RPA complex and RAD51. Ubiquitination of RFWD3

substrates has been proposed to temporally regulate the presence and activity of multiple effectors at stalled or collapsed replication forks[65]. Our study revealed that RFWD3 recruitment to chromatin is greatly diminished in SLFN11-expressing cells. Importantly, the ability of RFWD3 to interact with different chromatin-bound effector proteins, including chromatin remodelers, DNA repair, SUMOylation, and tRNA processing enzymes, are severely compromised in the presence of SLFN11 expression following HU-induced replication stress (Fig. 5C, Supplementary Fig. 7A). These results are consistent with a model in which SLFN11 restricts RFWD3's access to chromatin and/or replication fork and limits its engagement with replication stress response pathways. These findings align with emerging evidence that SLFN11 modulates protein ubiquitination at stalled forks under replication stress, potentially by altering the protein stability and activity of multiple ubiquitin ligases at the stalled fork[31].

Our results show that RFWD3 is required for PRIMPOL's localization at restarted forks in response to replication stress. PRIMPOL has previously been shown to be positively regulated by CHK1 phosphorylation and by interaction between its zinc finger domain and RPA bound to single-stranded DNA[66]. We speculate that robust activation of PRIMPOL may only be achieved under conditions of replication stress in which the ATR-CHK1 axis is activated and exclusively to forks enriched in Ub-RPA mediated by ATR-RFWD3. It is known that CHK1 negatively regulates dormant origin firing while positively regulating repriming by PRIMPOL through phosphorylation at Ser255[66]. It will be intriguing to determine whether CHK1 or other ATR downstream regulators co-regulate PRIMPOL activity for fork restart and recovery to help counteract sensitivity to chemotherapeutic agents in the absence of SLFN11. Such a global-local regulatory circuit would restrict PRIMPOL only to conditions under which global replication stress is high and its activity is limited locally to individual stalled forks.

Single-stranded DNA (ssDNA) gaps are a normal feature of every replication fork in the form of Okazaki fragments due to the discontinuous nature of lagging-strand replication[67]. Gaps can also occur when forks need to negotiate natural or unnatural DNA lesions or due to the presence of stable DNA secondary structures. For instance, gaps formed on the leading strand through the activity of PRIMPOL, which can prime at multiple locations along an ssDNA strand, leave smaller gaps that must subsequently be filled or processed. Prolonged exposure

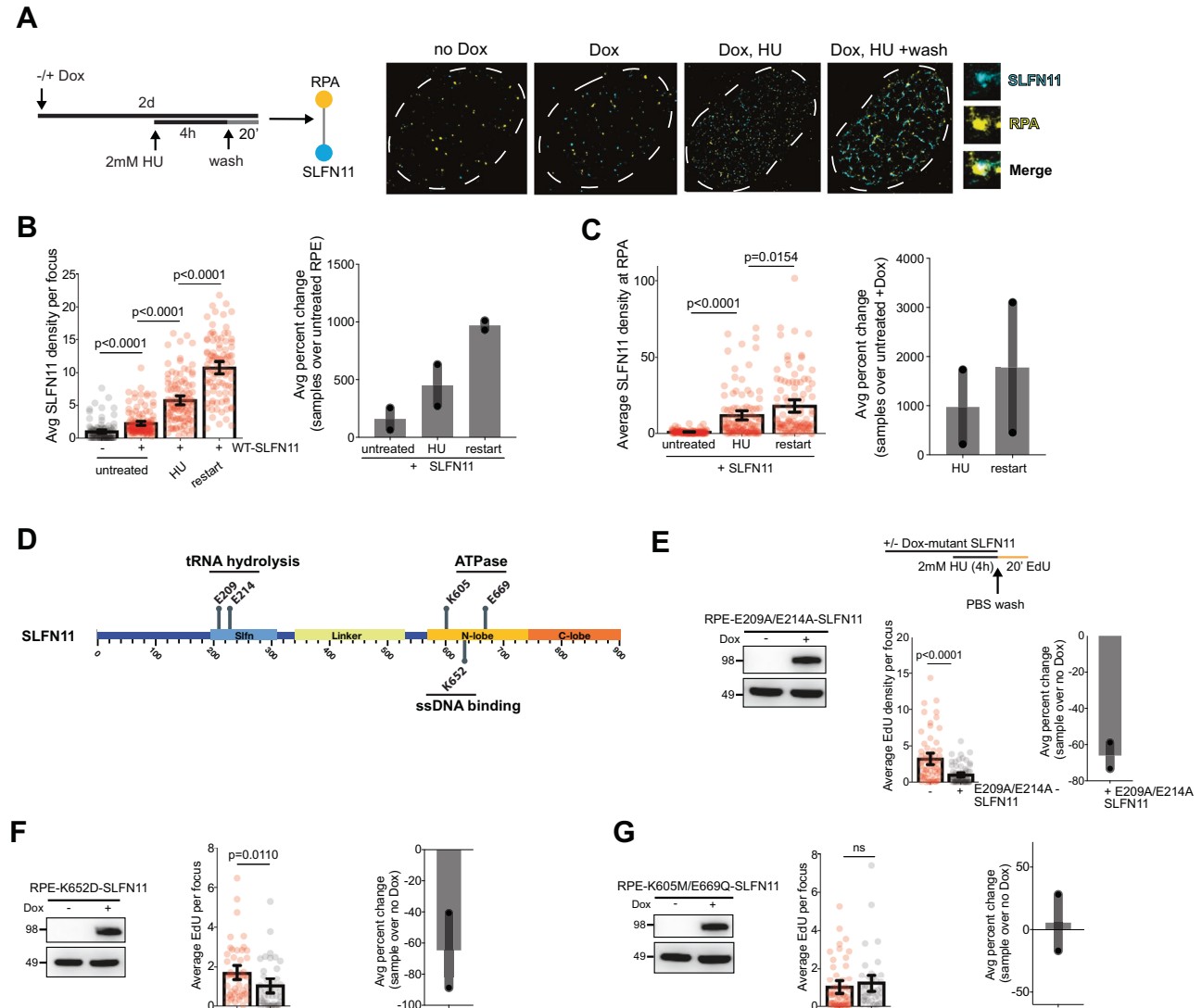

**Fig. 4 | SLFN11 persistently localizes with RPA at stalled replication forks. A**
(Schematic) -/+ Dox RPEs were treated with 2 mM HU for 4 h to stall forks and
released into fresh media to allow recovery (see schematic). Cells were stained with
SLFN11 (cyan) and RPA (yellow) prior to and after HU treatment, and after release
into fresh media for 20 min (Right). **B** (Left) Scatterplot quantification measuring the
average SLFN11 density per focus in PCNA+ nuclei. *p*-values of technical replicates
calculated using multiple unpaired two-tailed *t*-test from two biological replicates
(No Dox: *N* = 107; Dox: *N* = 130; Dox, HU: *N* = 96; Dox, restart: *N* = 95). Error bars =
mean, SEM. (Right) Bar graph showing the percent change of between the average of
two biological replicates, each normalized to the lowest mean condition relative to
no Dox-RPEs. **C** (Left) Scatterplot quantification measuring the average SLFN11
density at RPA per focus in PCNA+ nuclei using pair correlation analysis. *p*-values of
technical replicates calculated using multiple unpaired two-tailed *t*-test from two
biological replicates (Dox, no HU: *N* = 104; Dox, HU: *N* = 106; Dox, restart: *N* = 95).
Error bars = mean, SEM. (Right) Bar graph showing the percent change of between
the average of two biological replicates, each normalized to the lowest mean con-
dition relative to Dox RPEs with no HU. **D** Schematic of previously described SLFN11
functional domains and relevant amino acid residues: E209 and E214 (endonuclease),
K605 and E669 (ATPase), and K652 (single-stranded DNA binding). Dox-inducible
expression of mutant SLFN11 constructs containing point mutations in each of these
domains was generated in RPE-1 cells using lentiviral transduction (see "Methods").

**E** Whole cell lysates of E209A/E214A-SLFN11 RPEs were analyzed by Western blot for
SLFN11 (top) and alpha-tubulin (bottom). Schematic of mutant SLFN11 RPEs treated
with 2 mM HU for 4 h, followed by release into fresh media with EdU after washout to
measure fork restart. (Left) Scatterplot quantification measuring the average EdU
density per focus in PCNA+ nuclei. *p*-values of technical replicates calculated using
an unpaired two-tailed *t*-test from two biological replicates (no Dox: *N* = 65, Dox:
*N* = 73). Error bars = mean, SEM. (Right) Bar graph showing the percent change of
between the average of two biological replicates, each normalized to the lowest
mean condition relative to no Dox. **F** Whole cell lysates of K652D-SLFN11 RPEs were
analyzed by Western blot as in (**E**). (Left) Scatterplot quantification measuring the
average EdU density per focus in PCNA+ nuclei. *p*-values of technical replicates
calculated using an unpaired two-tailed *t*-test from two biological replicates (no Dox:
*N* = 51, Dox: *N* = 43). Error bars = mean, SEM. (Right) Bar graph showing the percent
change of between the average of two biological replicates, each normalized to the
lowest mean condition relative to no Dox. **G** Whole cell lysates of K605M/E669Q-
SLFN11 RPEs were analyzed by Western blot as in (**E**). (Left) Scatterplot quantification
measuring the average EdU density per focus in PCNA+ nuclei. *p*-values of technical
replicates calculated using an unpaired two-tailed *t*-test from two biological repli-
cates (no Dox: *N* = 55, Dox: *N* = 42). Error bars = mean, SEM. (Right) Bar graph
showing the percent change of between the average of two biological replicates,
each normalized to the lowest mean condition relative to no Dox.

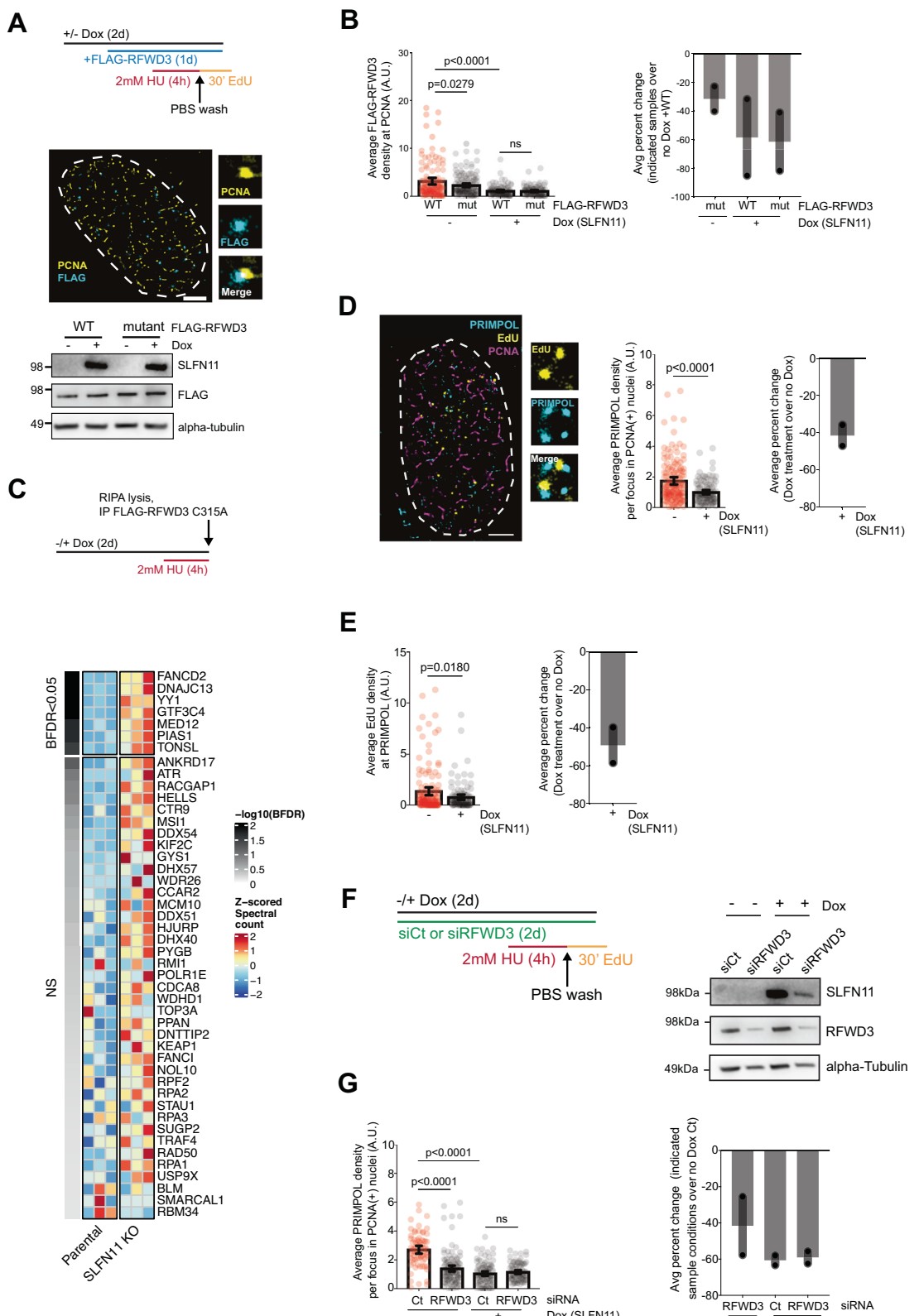

to these gaps can result in DNA breaks, leading to genomic instability. Therefore, these gaps must be processed by gap-filling mechanisms to complete DNA replication in a timely manner[68,69]. Recent studies have shown an association between the accumulation of ssDNA gaps generated by PRIMPOL and sensitivity to CPT and PARPi[70–74]. The presence of persistent ssDNA gaps has been attributed to defects in the FA pathway and backup Okazaki fragment processing[54,75–79].

Given that SLFN11 is a key driver of sensitivity to a variety of DNA-damaging chemotherapies, the identification of a downstream pathway with diminished activity presents an opportunity to explore novel therapeutic strategies. Several experimental small molecule inhibitors have been developed to target components of the FA pathway or adjacent processes, aiming to enhance sensitivity to DNA crosslinking agents such as cisplatin[80,81]. For example, USP1[82], FANCM and FANCL

**Fig. 5 | SLFN11 inhibits RFWD3 and PRIMPOL recruitment to restarted forks.**
**A** (Schematic) -/+ DOX-treated RPEs were transiently transfected with WT- or mutant-RFWD3-FLAG for 24 h. (Left) Representative SMLM image of RPEs stained with PCNA (yellow) and FLAG (cyan) following fork-stalling with 2 mM HU, and release into fresh media with EdU following washout. Scale bar = 2 μm. (Right) Whole cell lysates were analyzed by Western blot for SLFN11, FLAG, and alpha-tubulin (loading control). **B** Scatterplot quantification measuring pair-correlation analysis of the average WT- and mutant-RFWD3-FLAG density at PCNA in untreated and SLFN11-overexpressed RPEs. *p*-values of technical replicates calculated using an unpaired two-tailed *t*-test from two biological replicates (No Dox, WT: *N* = 127; No Dox, mutant: *N* = 135; Dox, WT: *N* = 100; Dox, mutant: *N* = 98). Error bars = mean, SEM. (Right) Bar graph showing the percent change of between the average of two biological replicates, each normalized to the lowest mean condition relative to no Dox + WT. **C** (Schematic) HAP1 parental and SLFN11 KO cells were treated with Dox for 24 h to induce stable expression of mutant-RFWD3-FLAG (C315A), followed by 2 mM HU treatment for 4 h. Heatmap showing Z-scored spectral counts for proteins previously identified as RFWD3 interactors by Yates et al[63]. **D** Representative SMLM image of RPEs stained with EdU (yellow), PCNA (magenta), and PRIMPOL (cyan) following fork-stalling with 2 mM HU, and release into fresh media with EdU following washout. Scale bar = 2 μm. (Left) Scatterplot quantification measuring the

average PRIMPOL density per focus in PCNA+ nuclei in untreated and SLFN11-expressed RPEs. *p*-values of technical replicates calculated using an unpaired two-tailed *t*-test from two biological replicates (Unt: *N* = 138, DOX: *N* = 108). Error bars = mean, SEM. (Right) Bar graph showing the percent change of between the average of two biological replicates, each normalized to the lowest mean condition relative to untreated RPEs. **E** (Left) Scatterplot quantification measuring the average EdU density PRIMPOL in untreated and SLFN11-expressed RPEs. *p*-values of technical replicates calculated using an unpaired two-tailed *t*-test from two biological replicates (Unt: *N* = 138, Dox: *N* = 108). Error bars = mean, SEM. (Right) Bar graph showing the percent change of between the average of two biological replicates, each normalized to the lowest mean condition relative to untreated RPEs. **F** (Schematic) -/+ DOX-treated RPEs were transfected with control siRNA (siCt) or siRFWD3. Whole cell lysates were analyzed by Western blot for SLFN11, RFWD3, and alpha-tubulin. **G** Scatterplot quantification measuring the average PRIMPOL density per focus in PCNA+ nuclei. *p*-values of technical replicates calculated using multiple unpaired two-tailed *t*-test from two biological replicates (siCt: *N* = 73, siRFWD3 = 117, DOX: *N* = 100, DOX/siRFWD3 = 87). Error bars = mean, SEM. (Right) Bar graph showing the percent change of between the average of two biological replicates, each normalized to the lowest mean condition relative to siCt-treated RPEs.

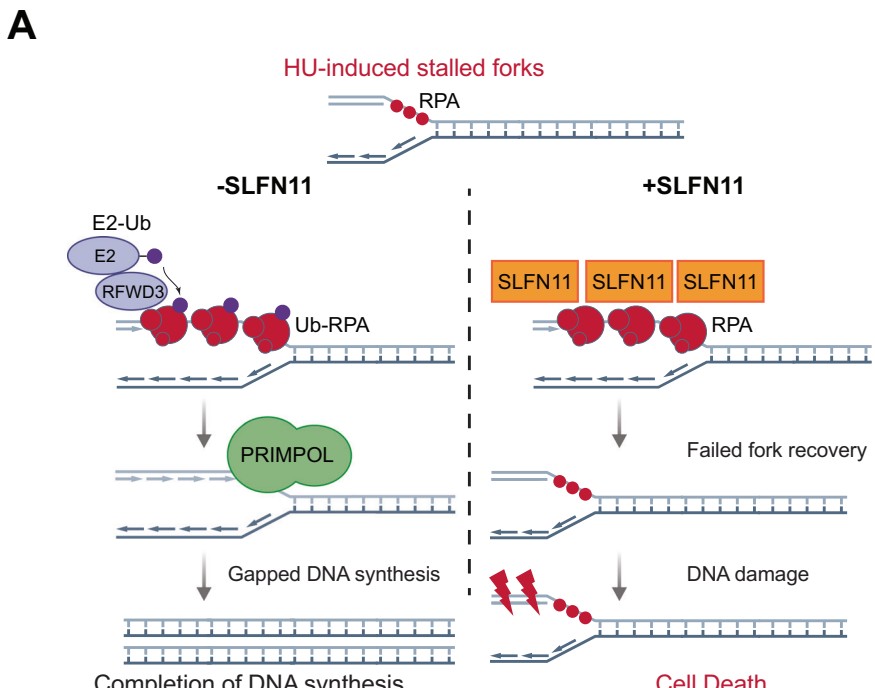

**Fig. 6 | Schematic model for the role of SLFN11 in counteracting the RFWD3-PRIMPOL DNA damage tolerance axis at stalled forks.** **A** In the absence of SLFN11, stalled replication forks accumulate ubiquitinated RPA through the replication stress-responsive ubiquitin E3 ligase RFWD3, allowing for PRIMPOL-

mediated gapped DNA synthesis and stalled fork recovery. In SLFN11-expressing cells, SLFN11 directly binds to RPA, thereby preventing the recruitment of both RFWD3 and PRIMPOL to chromatin/replication forks, leading to failed fork recovery, elevated replication-associated DNA damage, and cell death.

small molecule inhibitors have been investigated as a way to disrupt FANCD2/I activation[83,84], while inhibitors of ERCC1/XPF[85,86], REV7/REV3L[87], and RAD51[88–92] have been explored to inhibit lesion unhooking, TLS, and HR processes, respectively. Mutations in the WD repeat domain of RFWD3/FANCW disrupt binding to the RPA32 winged helix domain, leading to loss of substrate ubiquitination and a functional ICL repair defect[51]. Small molecules capable of dusripting WD-repeat domain interactions with key binding partners have been developed for other proteins, suggesting that the RFWD3-RPA32 interaction could be a tractable target to sensitize cancers to cisplatin[93]. PRIMPOL is also attracting attention as a new target for cancer therapy[94].

 Beyond the potential therapeutic implications, many fundamental questions remain unresolved. For instance, it is unclear

whether SLFN11 directly competes with RFWD3 at forks, or instead alters the dynamic turnover of RFWD3 and its substrates. Additionally, beyond RPA, which other RFWD3 ubiquitination substrates might be functionally critical for PRIMPOL-mediated fork recovery or HR-mediated repair remains unknown. Furthermore, while the study focuses on fork restart, RFWD3 is known to influence homologous recombination dynamics and checkpoint activation[33,34,50,51,53], so the relative contribution of RFWD3 antagonism to sensitivity across different lesion types, such as interstrand crosslinks, topoisomerase-induced breaks, replication-blocking agents, or PARP inhibitors, remains to be determined. Finally, whether SLFN11 might affect downstream gap-filling pathways also represents a significant unanswered question.

Although this study shows how SLFN11 disrupts recovery of stalled replication forks via antagonism of RFWD3–PRIMPOL, SLFN11 is also known to exert p53-independent apoptotic effects by depleting specific tRNAs and impairing ribosome biogenesis[26]. A crucial open question is whether these mechanisms are functionally separable or inherently interconnected in time or by mechanism. For instance, is the dynamic recruitment of SLFN11 to stressed forks required to trigger secondary ribotoxic stress and tRNA cleavage? Or are its tRNA endoribonuclease activity and fork binding distinctly regulated? Our findings using separation of function mutants show that while ssDNA binding, tRNA hydrolysis, and intact ATPase domain are all required for HU sensitivity, only ATPase mutants were defective in persistence at replication forks undergoing restart. This prompts the question: What is the independent contribution of fork blockage versus translational inhibition to cell death and sensitivity to different agents? Understanding how SLFN11's fork-based inhibition and tRNA-mediated apoptosis influence each other could reveal which mechanism is dominant in different therapeutic contexts.

The present study reveals an RFWD3-dependent role for PRIMPOL in promoting efficient fork restart in cells that do not express SLFN11. Despite PRIMPOL's critical role in fork restart to minimize replication-associated DNA damage, ssDNA gaps are still likely to accumulate behind the restarted forks, and it isn't clear how these ssDNA gaps will eventually be filled or whether gap-filling processes will have negative consequences for genome maintenance. As PRIMPOL localization and function at the fork is regulated by RFWD3, and with RFWD3 being an FA gene, it is intriguing in future studies to assess how or whether RFWD3-mediated RPA ubiquitination, and potentially of other targets, is mechanistically linked to fork restart through the activity of PRIMPOL.

## Methods
### Key resources and reagents
Key resources and reagents used in this study are described in Supplementary Table 1.

### Cell culture and cell lines
The HAP1 cell line is a near-haploid human cell line derived from the KBM-7 chronic myelogenous leukemia (CML) cell line (Horizon Discovery). HAP1 cells were cultured at 37 °C in IMDM media (Gibco) supplemented with 10% FBS (Atlantic Biologicals). HAP1 cells expressing the pLIX403-SLFN11 doxyciclin-inducible expression construct were cultured in IMDM media with 10% tetracycline negative FBS and 1 mM puromycin; SLFN11 expression was induced by the addition of 1 µg/ml doxycycline (Fisher Scientific) to the media for 48 h. All other cell lines were obtained from ATCC (American Type Culture Collection). U2OS cells were cultured in DMEM media (Gibco) supplemented with 10% FBS, 1% penicillin/streptomycin (Gibco) and 1% glutamine. RPE-1 cells were grown in DMEM/F12 (Gibco) with 10% FBS, 1% penicillin/streptomycin, and 0.25% Sodium Bicarbonate (Gibco). SW1271 and DU145 cells were cultured in RPMI (Gibco) with 10% FBS.

### CRISPR gene disruption and lentivirus production
HAP1 cells were electroporated with Cas9 ribonucleoprotein complexes targeting coding exons 1, 3, and 4 of Schlafen family member 11 (SLFN11) using the Neon Transfection System (Thermo Fisher Scientific). Cells were allowed to recover in complete medium and expand before they were single-cell sorted into 96-well tissue culture plates by fluorescence-activated cell sorting. Multiple individual clones for each cell line were expanded into 6-well plates, reserving a portion for cryopreservation. The 6-well plates were screened for clones harboring frameshift indels in the SLFN11 coding sequence using PCR amplification and Sanger sequencing across the targeted locus. Loss of SLFN11 protein expression was confirmed by immunoblotting. SLFN11 cDNA was cloned directly from parental HAP1 cells by polymerase chain reaction into the pDONR221 plasmid using Gateway cloning (Thermo Fisher). SLFN11 cDNA was subcloned from pDONR221 into the doxycycline inducible pLIX403 lentiviral transfer vector. Correct cloning outcomes were confirmed by whole-plasmid sequencing. pLIX403-SLFN11 transfer vector (3 µg) was transfected into HEK293T/17 cells in T25 flasks along with lentiviral packaging vectors psPAX2 (2 µg) and pMD2.G (1 µg) using 12 µg polyethylenimine. The following day media was changed and the cells were incubated a further 48 h. Lentiviral supernatants were collected and clarified by centrifugation at 300 g for 5' followed by syringe filtration using 0.45 µm pore size (Millipore). Between 100 and 1 µL of clarified lentiviral supernatants were added to SLFN11 KO cells in 6-well plates in the presence of 8 µg/mL polybrene. After 48 h, the media were replaced with fresh media containing a final concentration of 1 µg/mL puromycin. To favor selection of cells with a single integrated provirus, the well with surviving cells to which the least amount of viral supernatant was added was expanded as a stable cell line.

### DNA fiber analysis
For analysis of replication fork restart, cells were pulse-labeled with 50 µM IdU for 20 min, washed twice with PBS, and treated with 2 mM hydroxyurea (HU) for 4 h to stall replication through nucleotide depletion. Replication was then allowed to resume by washing out HU with PBS and adding back fresh media with 50 µM CldU for 30 min. For experiments involving S1 nuclease digestion, following the CldU pulse, cells were treated with CSK100 buffer (100 mM NaCl, 10 mM HEPES, 3 mM MgCl$_2$ [pH 7.2], 300 mM sucrose, and 0.5% Triton X-100) for 10 min at room temperature, then incubated with S1 nuclease buffer (30 mM sodium acetate [pH 4.6], 10 mM zinc acetate, 5% glycerol, and 50 mM NaCl) with or without 20 U/mL S1 nuclease (Thermo Scientific, EN0321) for 30 min at 37 °C. Cells were then collected, washed and resuspended in cold 1X PBS at a density of $1 \times 10^6$ cells/ml before spotting on a glass slide. Cells were lysed in SDS lysis buffer [0.5% SDS, 200 mM Tris-HCl (pH 7.4), 50 mM EDTA] for 6 min prior to tilting at a 15° angle to allow DNA spreading. Stretched DNA fibers were fixed in a chilled solution of methanol: acetic acid (3:1) for 3 min, denatured with 2.5 N HCl for 30 min, washed in 1X PBS, and incubated in blocking buffer (5% BSA in PBS with 0.1% Triton X-100) for 1 h at room temperature. Slides were probed with primary antibodies (mouse anti-IdU [B44] (BD Biosciences 347580, 1:150 dilution) and rat anti-CldU [BU1/75 (ICR1)] (Abcam ab6326, 1:200 dilution)) and then with fluorescent-dye conjugated secondary antibodies (goat anti-mouse IgG (H + L) Alexa Fluor 488 (Thermo Fisher A11001, 1:350 dilution) and goat anti-rat IgG (H + L) Alexa Fluor 594 (Thermo Fisher A11007, 1:350 dilution)). Images were collected using a Keyence BZ-X710 microscope. Fork speed following restart (CldU tract length) was assessed in a minimum of 180 fibers for each independent experiment, and the analysis shows the pool of three biological replicates per condition. A minimum of 250 DNA fibers were measured for each independent experiment of percentage of restart (% stalled forks), and analysis shows mean for three independent experiments. Tract lengths were determined in ImageJ using the scale of 1 µm = 2.59 kb.

### Western blot analysis, siRNA and cDNA transfections
For routine western blotting analysis, cells were lysed in denaturing SDS buffer (100 mM Tris (pH 6.8), 2% SDS and 20 mMγ-mercaptoethanol), and cell extracts were separated on NuPAGE 4–12% Bis-Tris or 3–8% Tris-Acetate gels (Invitrogen). Proteins were transferred onto 0.45 µm PVDF membrane (Millipore) in Tris-Glycine transfer buffer (Invitrogen). Membranes were blocked in 5% milk in TBS-T for 1 h prior to incubation with primary antibody overnight. The following day, membranes were incubated with HRP-conjugated secondary antibodies (Peroxidase AffiniPure Goat Anti-Mouse IgG (H + L) (Jackson Labs 115-035-003, 1:10,000 dilution) and Peroxidase AffiniPure Goat Anti-Rabbit IgG (H + L) (Jackson Labs 111-035-003, 1:10,000 dilution))

in 5% milk in TBS-T and developed using ECL Prime reagent (GE Healthcare). Transient siRNA transfection complexes were performed using Lipofectamine RNAimax (Invitrogen) reagents in Opti-MEM media (Gibco, 31985062) according to the manufacturer's instructions in penicillin/streptomycin-free media. Downstream analyses were performed after 72 h of siRNA knockdown. siRNA sequences can be found in the Key Resource Table. WT-RFWD3-FLAG plasmid (see Key Resource Table) complexes were transiently expressed for 24 h using Fugene HD (Promega, E2311) and Opti-MEM media in penicillin/strep-tomycin-free media according to manufacturer's instructions.

### Immunofluorescence staining

For detection of 53bp1 nuclear bodies, cells plated on glass coverslips were fixed for 15 min in 4% paraformaldehyde followed by 5 min in ice-cold methanol. Cells were permeabilized in 0.5% Triton X-100 buffer for 10 min, blocked for 1 h in 2% BSA, 0.2% Trixon X-100 buffer, and incubated for 2 h with primary antibodies (rabbit anti-53BP1 (Abcam ab175933, 1:200 dilution) and mouse anti-Cyclin A2 (Calbiochem CC17, 1:100 dilution)) at room temperature. Coverslips were then washed in PBS, incubated for 45 min in secondary antibodies (Goat anti-mouse IgG (H + L) Alexa Fluor 546(Thermo Fisher A11003, 1:350 dilution) and Goat anti-rabbit IgG (H + L) Alexa Fluor 488 (Thermo Fisher A11008, 1:350 dilution)), washed again with PBS, and mounted onto glass slides using Prolong Gold Antifade mountant with DAPI (Thermo Fisher Scientific). Imaging was performed using a Keyence BZ-X710 micro-scope with a 40X objective. A minimum of 200 Cyclin A-negative cells were analyzed for 53BP1 foci from three independent experiments.

### Multi-color single-molecule super-resolution microscopy (SMLM)-STORM

SMLM and downstream analysis was performed as previously described[95–98]. Briefly, RPE-1 cells were treated with 2 mM HU for 4 h, washed with PBS to remove HU, released into fresh media for 10 min, then labeled with 10 μM EdU for 30 min at 37 °C. Cells were CSK extracted (10 mM HEPES pH=7.4, 300 mM sucrose, 100 mM NaCl, 3 mM MgCl$_2$, 0.5% v/v Triton X-100) for 10 min, gently washed with PBS, and fixed in 4% v/v paraformaldehyde in PBS for 15 min at room temperature. Cells were then blocked in buffer (2% w/v BSA, 2% w/v glycine, 0.2% w/v gelatin, 50 mM NH$_4$Cl) for 15 min at room tempera-ture or overnight at 4 °C. EdU was detected by Click-It chemistry according to manufacturer's instructions (Thermo Fisher), and cov-erslips were stained with antibodies against targets of interest the day of imaging: mouse monoclonal anti-PCNA (sc-56) at 1/1000 in blocking buffer followed by secondary staining of with goat anti-mouse IgG AF568, AF488-conjugated-gH2AX (pSer139) (Millipore, AF488 05-636) at 1/7500 in blocking buffer, anti-FLAG-M2 (Sigma, F1804) diluted 1/1000 in blocking buffer followed by secondary staining with goat anti-mouse AF488 diluted in blocking buffer at 1/10000, and anti-PRIMPOL (Proteintech, 29824-1-AP) diluted 1/2000 in blocking buffer followed by secondary staining with goat anti-rabbit AF488 diluted 1/10000 in blocking buffer (see Key Resources Table). Coverslips were mounted on imaging slides and washed with imaging buffer (1 mg/mL glucose oxidase, 0.02 mg/mL catalase, 10% glucose, 100 mM cystea-mine) prior to imaging.

SMLM imaging was accomplished on a custom-built inverted microscope based on the ASI RAMM platform: a 561 nm laser (Coher-ent, Sapphire 561 LPX-500) and a 639 nm laser (ultralasers, MRL-FN-639-800) were aligned, expanded, collimated, and directed into a TIRF objective (Olympus, UApo N, 100x NA1.49) using a penta-edged dichroic beam splitter (Semrock, FF408/504/581/667/762-Di01). Lasers were adjusted to Highly Inclined and Laminated Optical sheet (HILO) mode prior to imaging with the illumination intensity at ~1.5 kW/cm$^2$ for the 639 nm laser and and 1.0 kW/cm$^2$ for the 561 nm laser at the exit of the objective. A 405 nm laser (ultralasers, MDL-III-405-100) was used to drive AF647 to its ground state. AF647 was

illuminated by the 639 nm laser, then JF549 was illuminated by the 561 laser. Both of their fluorescence was expanded to 1.67x and filtered by a single-band pass filter (Semrock FF01-676/37 for AF647, and FF01-607/36 for JF549). A sCMOS camera (Teledyne Photometrics, Prime 95B) was used for collection at 33 Hz / 30 ms per frame for a minimum of 2000 frames for each channel.

A regular DAOSTORM routine was used for single-molecule localization, as previously described[95,98]. Frames were individually stacked with a smaller (σ- = 143 nm) and larger (σ- = 286 nm) Gaussian kernel, and local maxima were identified by subtracted the bigger from the smaller filtered images. A 9 × 9 square (1 pixel is about 65 nm) around each local maximum was cropped and submitted for MFA sub-pixel localization by fitting the data to one or more 2D-Gaussian point spread functions through maximum likelihood estimations (MLE). The fitting accuracy of each target of interest that was given by the Cramér-Rao Lower Bound was analyzed by fitting its distribution with a skew-Gaussian distribution, using the center as the average localization precision. Localizations that appeared within 2.5 times of the average localization in consecutive frames were averaged and considered as a localization from one blinking event. The coordinates of such events were then submitted for pair-correlation analysis. Representative SMLM images were generated from the coordinate list to the 10 nm pixel canvas and blurred with a Gaussian kernel (σ ~ = 143 nm) for display.

SMLM data analyses, including the alignment of three-color channels, Auto-Pair Correlation (PC), and Cross-PC analysis, were performed as previously described[95–98]. Briefly, pair-wise distances between every molecule of one color to all the molecules in a different color are computed directly from the molecular coordinates. Based on all the measured pair-wise distances, the algorithm then automatically generates the probability distribution of the distances (probably density) using a cross-correlation (or pair-correlation) function. Rele-vant SMLM scripts can be found at: https://github.com/yiny02/direct-Triple-Correlation-Algorithm.

### Immunoprecipitation and mass spectrometry (IP/MS)

Stable HAP1 cells expressing doxycycline-inducible Flag-hRFWD3-C315A were generated by lentiviral transduction and selected with 1 μg/mL puromycin. Cells were seeded in 10 cm dishes and treated with 1 μg/mL doxycycline for 24 h to induce expression, followed by treat-ment with or without 2 mM hydroxyurea (HU) for 4 h. After treatment, cells were washed three times with cold PBS and lysed in ice-cold RIPA buffer (Thermo Fisher, 89901) supplemented with protease inhibitors (Thermo Fisher, 78437), phosphatase inhibitors (Thermo Fisher, 78420), and benzonase nuclease (Sigma, E1014). Lysates were incu-bated overnight at 4 °C with anti-FLAG M2 affinity gel (Sigma, A2220) on a rotating platform. Beads were washed thoroughly with cold RIPA buffer, and bound proteins were eluted using 1.5 mg/mL 3X DYKDDDDK peptide (Thermo Fisher) resuspended in PBS. Each experimental condition was performed in biological triplicate.

For mass spectrometry analysis, eluates were resolved briefly by SDS-PAGE and visualized using GelCode Blue Safe Protein Stain (Thermo Fisher, 1860957) to confirm protein recovery. Samples were reduced with 3 μL of 0.3 M dithiothreitol (Sigma-Aldrich) and subse-quently alkylated with 3 μL of 0.5 M iodoacetamide (Sigma-Aldrich) for 45 min at room temperature in the dark. Each sample was divided into three aliquots for digestion with different enzymes (trypsin, Glu-C, and chymotrypsin (Sigma-Aldrich) to improve sequence coverage of hRFWD3-C315A. To each aliquot, 250 ng of SP3 beads were added, and proteins were precipitated onto the beads by adding 100% ethanol. Beads were washed four times with 80% ethanol to remove excess flag tag peptide and/or detergents. Separate aliquots were digested with 500 ng of sequencing-grade trypsin (Promega) in 100 mM ammonium bicarbonate, 1 μg of Glu-C (Sigma-Aldrich) in 1X PBS buffer, or 1 μg of chymotrypsin in 100 mM ammonium bicarbonate. Trypsin and Glu-C

digestions were carried out overnight on a shaker at 800 rpm at room temperature, while chymotrypsin digestion was quenched after 2 h to prevent over digestion. Supernatants were transferred to new vials and acidified to pH ~2 using 5% formic acid (FA). Acidified peptides were loaded onto EvoTips spiked with synthetic peptides, iRT (Biognosys), for LC-MS analysis.

Peptide separation was performed online using an EvosepOne LC system equipped with a PepSep C18 analytical column (15 cm long, 150 μm ID, 1.9 μm beads, cat# EV-1106). Peptides were eluted over an 88-min gradient at a flow rate of 220 nL/min and introduced directly into a timsTOF HT mass spectrometer operated in data-dependent acquisition (DDA)-PASEF mode. Mass spectra were recorded over an m/z range of 100–1700. Ion mobility separation was performed from 0.60 to 1.60 V s/cm² over a 100 ms ramp time. MS/MS spectra were acquired using 10 PASEF scans per cycle with a polygon filter to exclude singly charged ions. Ions within the polygon region were subject to an intensity threshold of 2500, and the target intensity for MS/MS acquisition was set to 20000. A dynamic exclusion of 0.4 min was applied. Collision energies were scaled linearly from 20 to 60 eV for ions with inverse reduced mobilities (1/K0) ranging from 0.60 to 1.60 V s/cm².

To quantify phosphorylation sites on hRFWD3 C315A, an additional aliquot of the trypsin-digested samples was analyzed using data-independent acquisition (DIA)-PASEF mode. DIA acquisition was performed using 22 mass windows with variable isolation widths within the polygon region. All other acquisition parameters were kept consistent with the DDA acquisitions. The corresponding mass spectrometry data have been deposited in the MassIVE public repository (UCSD) under the accession ID MSV000098536.

To identify potential binding partners, MS/MS spectra from DDA experiments were searched using Byonic (Protein Metrics) against the UniProt human database. Search parameters included fixed carbamidomethylation of cysteines and dynamic modifications for methionine oxidation, phosphorylation (on serine, threonine, and tyrosine), and deamidation (on asparagine and glutamine). Enzyme specificity was set to specific for trypsin and Glu-C, and semi-specific for chymotrypsin, allowing up to two missed cleavages. Peptide spectrum matches (PSMs) were filtered to achieve a false discovery rate (FDR) of less than 1%, and only proteins identified with two or more unique peptides were considered. Differential protein enrichment between parental and knockout cell lines was assessed using SAINT scoring with a 5% FDR threshold.

The differentially enriched proteins were subjected to pathway overrepresentation analysis in R v4.4.2. First, the canonical UniProt accession numbers were converted to Entrez IDs using the bitr function, followed by KEGG and Reactome pathway enrichments using the enrichKEGG and enrichPathway functions from the clusterProfiler R package v4.14.4. The background protein list was set to the list of proteins that underwent differential enrichment testing using SAINT. The protein-protein interaction network for the differentially enriched proteins was constructed using the stringApp v2.2.0 plugin within Cytoscape v3.10.3.

### Ubiquitinated RPA32 pulldown and detection

Stable HEK293T cells expressing doxycycline-inducible SLFN11 were generated by lentiviral transduction and selected with 1 μg/mL puromycin. Cells were seeded in 10 cm dishes and transfected with the pSAM504-FLAG-Ub vector using X-tremeGENE™ HP DNA Transfection Reagent (Sigma, 6366236001). After 24 h, the media was replaced with fresh media containing ±1 μg/mL doxycycline to induce SLFN11 expression. Following an additional 24-h induction period, cells were treated with or without 2 mM hydroxyurea (HU) for 4 h. Cells were then washed three times with cold PBS and lysed in ice-cold RIPA buffer (Thermo Fisher, 89901) supplemented with protease inhibitors (Thermo Fisher, 78437), phosphatase inhibitors (Thermo Fisher,

78420), and benzonase nuclease (Sigma, E1014). Lysates were incubated overnight at 4 °C with anti-FLAG M2 affinity gel (Sigma, A2220) on a rotating platform. Beads were washed thoroughly with cold RIPA buffer, and bound proteins were eluted using 1.5 mg/mL 3X DYKDDDDK peptide (Thermo Fisher) resuspended in PBS. Eluates were subjected to SDS-PAGE and western blotting. Membranes were probed with anti-RPA32 antibody (Cell Signaling Technology, 52448) diluted in 5% milk in TBS-T, followed by HRP-conjugated secondary antibody (Thermo Fisher, 31460). Signal was detected using SuperSignal West Femto Chemiluminescent Substrate (Thermo Fisher, 34580) and imaged using an iBright Imaging System (Thermo Fisher).

### EdU incorporation and cell cycle analysis

For cell cycle analysis, HAP1 cells were pulse-labeled with 10 μM 5-ethynyl-2′-deoxyuridine (EdU) for 1 h at 37 °C, then washed once with PBS and harvested by trypsinization. Cells were fixed in 3.7% formaldehyde, permeabilized with 0.5% Triton X-100 in PBS, and treated with 100 μg/mL RNase A (New England Biolabs, T3018L) for 30 min at 37 °C. EdU was labeled using the Click-&-Go® Plus EdU 647 Flow Cytometry Assay Kit (Click Chemistry Tools, Cat. 1381) according to the manufacturer's protocol. Cells were stained with 1 μg/mL DAPI for 5 min at room temperature, washed twice with PBS, and resuspended in PBS for analysis. Flow cytometry was performed using a BD FACSymphony A5 flow cytometer. Data were analyzed using FlowJo software (BD Biosciences).

### Cell viability assay

HAP1 cells were plated into Corning® 96 Half Area White Flat Bottom plates (Corning, 3688) and treated the following day with a range of hydroxyurea (HU) concentrations from 1.57 μM to 15,000 μM. After 48 h of treatment, cell viability was assessed using the CellTiter-Glo® 2.0 Cell Viability Assay (Promega, G9242) according to the manufacturer's instructions. Luminescence was measured using a BioTek Synergy H1 plate reader. Values were normalized to the untreated control, and $EC_{50}$ values were calculated using GraphPad Prism software.

### Statistics and reproducibility

Primary data were recorded using Microsoft Excel, and statistical analyses were performed using GraphPad Prism 8, FlowJo 10, ImageJ 1.52a, Matlab (v2017b), and OriginLab (2018) software. Exact p-values are provided for each experiment in the Source Data file associated with this article. All SR experiments were performed at least in duplicate with $n > 60$ sample size. All IF experiments, including DNA fiber analyses, are based on three biological replicates with a. sample size $n > 200$. Western blotting experiments were performed in at least three independent experiments. Source data contains all raw data used for statistical analysis. Representative experiments shown in all the figures and Supplementary Figs. are results of experiments that were done at least three times ($n = 3$) independently with similar results.

### Reporting summary

Further information on research design is available in the Nature Portfolio Reporting Summary linked to this article.

## Data availability

The mass spectrometry data have been deposited to the ProteomeXchange consortium via the MassIVE partner repository (UCSD) with dataset identifier PXD066219. [https://proteomecentral. proteomexchange.org/cgi/GetDataset?ID=PXD066219]. Source Data have been deposited in the database in Figureshare (https://doi.org/10. 6084/m9.figshare.30172414).

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

## Acknowledgments

We thank members of the Poirier, Huang, and Rothenberg labs for technical assistance and critical discussions. L.G. was supported by the NYU MSTP Scholar Award; T.T.H. by the generous gift from the Goldberg Family Foundation; E.R. and T.T.H. by the generous gift from Laura Chang and Arnold Chavkin. This work was supported by NIH grants: GM139610 and ES031658 (T.T.H.); P01CA288368 (T.T.H. & J.T.P.); R35 GM134947, AI153040, and CA247773 (E.R.). The mass spectrometry and cell sorting/flow cytometry technologies were provided by the Proteomics Laboratory (RRID: SCR_017926) and Cytometry and Cell Sorting Laboratory (RRID: SCR_019179) cores, respectively, at NYU Langone Health, which is supported in part by the NIH/NCI grant P30CA016087 at the Laura and Isaac Perlmutter Cancer Center.

## Author contributions

K.E.C., D-W.S., L.G., T.T.H., and J.T.P. conceived and designed the research project. K.E.C. performed the DNA fiber analysis. D-W.S. developed and validated key cell lines and reagents for the study, performed FLAG-Ub pulldown assays, and IP/MS experiments. L.G. performed the SMLM imaging studies. K.E.C., D-W.S., and L.G. performed the research and collected data. K.E.C., D-W.S., L.G., B.S., D.F., E.R., T.T.H., and J.T.P. analyzed and interpreted the data. K.E.C., D-W.S., L.G., T.T.H., and J.T.P. wrote the initial draft of the manuscript, and T.T.H. and J.T.P. edited/revised the manuscript.

## Competing interests

T.T.H. and J.T.P. have applied for a method of use patent related to targeting RFWD3 and PRIMPOL in cells that lack SLFN11. The remaining authors declare no competing interests.
