## [Transparent Peer Review file · Nature Communications]

SLFN11 counteracts the RFWD3-PRIMPOL DNA damage tolerance axis to restrain gapped DNA synthesis in response to replication stress

Corresponding Author: Professor Tony Huang

Version 0:

Reviewer comments:

Reviewer #1

(Remarks to the Author)

Reviewer's comments on 'SLFN11 counteracts the RFWD3-PRIMPOL DNA damage tolerance axis to inhibit DNA replication fork recovery' authored by Coleman et al.,

Replication stress is caused by the slowing or stalling of replication forks and can happen for many reasons, including DNA damage, replication-transcription conflicts, fork defects, dNTP imbalance etc. A stalled fork can be restarted by various mechanisms, including fork reversal, DNA repair pathways, and recruitment of factors that can stabilize and help re-initiation of DNA replication. In the current manuscript, the authors study SLFN11, a factor with tRNA endoribonuclease and DNA binding functions, and its role in replication fork dynamics, especially in response to replication stress. Using DNA fiber analysis, the authors propose that SLFN11 antagonizes a fork recovery pathway involving the E3 ligase, RFWD3, and the human DNA primase-polymerase (PRIMPOL). The data is clean and, for the most part, convincing. However, mechanistic details on how SLFN11 counteracts the RFWD3-PRIMPOL axis were missing. The entire manuscript is based on interesting observations from DNA fiber analysis, but none of the experiments get to the mechanism of action. The observations made in this manuscript are interesting but too preliminary.

Some specific comments are noted below:

- 1) Figure S1B: There is a noticeable difference in the EdU vs DAPI flow data between the control and SLFN11 KO. There is a population of cells between 2C and 4C content (with no excitation on the Y-axis), suggesting that replication is progressing at a very slow rate or cells are in the S-phase, but the time given for EdU incorporation is insufficient. Further, late S phase seems to be impacted. It is incorrect to state that there are no detectable changes.
- 2) For all the DNA fiber experiments, a thorough quantification of the IdU-only tracts must be presented. The length of CldU after HU would strengthen their argument about whether the replication upon restart was impacted or not.
- 3) Figure 1C: The finding that ATR signaling remains intact, but RPA32 ubiquitination is altered in SLFN11-expressing cells is intriguing. What is the potential mechanism linking SLFN11 to RFWD3-mediated ubiquitination? Does SLFN11 directly or indirectly influence RFWD3 activity? Is there any evidence for an interaction between SLFN11 and RFWD3 (based on IPs or Alpha-fold predictions)?
- 4) Figure 1D: the authors need to provide better quantifiable data. Is there a dose-dependent change to RPA Ub when SLFN11 is induced?
- 5) Figure 1F: Which band represents RFWD3? Is there an upregulation of RFWD3 in cells lacking SLFN11, and is this increase resulting in enhanced ubiquitination of RPA32?
- 6) Figure 1G: The authors demonstrate that RFWD3 is required for efficient fork restart in SLFN11 KO cells. However, it remains unclear whether RFWD3 also affects fork elongation rates post-restart. As mentioned in #2, quantification of fiber data is critical. Measuring elongation in RFWD3-depleted cells will help determine if its role is specific to restart or if its depletion causes a broader replication defect. If RFWD3 depletion also slows elongation, the observed effects may reflect a general impairment in replication progression rather than a restart-specific function.
- 7) The reviewer is not convinced with the following statement: Together, these results confirm that the replication fork restart defect in SLFN11-expressing cells is consistent with insufficient RFWD3 activity. Why is the red tract starting and then not continuing? Is it taking longer to restart, or does it only progress to a certain extent once it restarts? For example, 1G- how

many never fire after HU? And does that change upon the loss of SLFN11? And what happens when RFWD3 is co-depleted?

8) Another statement that needs clarification: Taken together, these data indicate that ssDNA gaps in SLFN11 knockout cells are likely dependent on PRIMPOL engagement downstream of RFWD3 function to restart replication forks. What is the proof that it is downstream of RFWD3? The defect could also be caused by a global defect in replication restart caused by RFWD3 depletion.

9) Figure S4C: Can a positive control be included?

10) Figure 3C-D: Are these images only representing early S-phase cells? How is the quantification in C and D done? What is EdU density? Or does this represent the number of EdU foci?

11) Figure 3E: What does the image provided represent? Is it a control cell or upon dox induction?

12) How is the extent of induction of SLFN11 controlled? Does the extent of induction impact primpol function?

13) Figure 3G: The reviewer is unconvinced about the image quality of gH2AX. Is this a single section? What is the resolution of SMLM? Why is PCNA not showing complete overlap with EdU, considering that PCNA can localize to replication and repair sites?

14) The data presented in the figure doesn't match what is mentioned in the text: SMLM analysis of PRIMPOL reveals that PRIMPOL recruitment to forks is compromised by knockdown of RFWD3, which is consistent with our findings that knockdown of PRIMPOL did not impact RFWD3-mediated RPA ubiquitination (Figure 4D).

15) The authors must provide additional compelling evidence confirming that PRIMPOL's role in fork restart depends on RFWD3. For instance, do stalled forks persist longer when RFWD3 and PRIMPOL are depleted compared to RFWD3 knockdown alone?

16) The proposed model is oversimplified and leaves unanswered questions regarding the precise mechanisms by which SLFN11 influences RFWD3 and PRIMPOL recruitment and function during replication fork restart. For instance, does SLFN11 directly inhibit RFWD3's E3 ligase activity, or how does it impact the localization of RFWD3 and PRIMPOL to the fork?

17) Minor comment: Within the introduction, the description of SLFN11's proposed functions based on previous literature is quite dense and lists multiple mechanisms in rapid succession. This section could be broken into chunks, and specifying which mechanisms are more relevant would provide clarity.

Also, RFWD3 and PRIMPOL have not been sufficiently introduced. Since they are central to the study, briefly describing their known roles in replication fork restart before discussing their connection to SLFN11 would improve context.

Reviewer #2

(Remarks to the Author)

In this manuscript, the authors demonstrate that SLFN11 expression impairs replication fork restart by inhibiting the RFWD3-PRIMPOL DNA damage tolerance axis, a pathway critical for resolving stalled replication forks, promoting replication stress recovery, and maintaining genomic stability. They further show that SLFN11 suppresses RFWD3-PRIMPOL localization at restarted forks and that PRIMPOL recruitment to replication forks is largely dependent on RFWD3.

While these findings are of potential interest to the field, the manuscript lacks mechanistic insights into how SLFN11 disrupts the RFWD3-PRIMPOL axis to inhibit replication fork recovery. Without a more detailed mechanistic characterization, the study remains preliminary. Therefore, I consider the manuscript insufficiently developed for publication in its current form.

Major Points

1. The proposed counteraction of the RFWD3-PRIMPOL axis by SLFN11 at stalled replication forks is intriguing; however, the precise molecular mechanism remains unclear. Since SLFN11, RFWD3, and PRIMPOL are all RPA-interacting proteins and are recruited to stalled forks via RPA, could SLFN11 compete with RFWD3 and PRIMPOL for RPA binding, thereby preventing their localization to stalled forks? If so, how is this competition regulated? Further investigation into the interplay between SLFN11, RFWD3, and PRIMPOL at RPA-coated forks is necessary to substantiate this model.

2. Although the authors demonstrate that RFWD3 is required for PRIMPOL recruitment to stalled forks, the underlying mechanism is not addressed. Does RFWD3 directly mediate PRIMPOL recruitment through physical interaction, post-translational modifications, or modulation of RPA dynamics? Additional experiments are needed to delineate this regulatory pathway.

3. In Figures S2D-E, the authors show that RFWD3 knockout (KO) clones derived from SLFN11 KO cell lines exhibit a phenotype similar to the HAP1 parental SLFN11-expressing cells. To determine whether RFWD3's E3 ligase activity is critical for this effect, the authors should perform complementation assays by reintroducing wild-type RFWD3 and an E3 ligase-inactive mutant into RFWD3 KO cells. This would clarify whether the observed phenotype is dependent on RFWD3-mediated ubiquitination.

Minor Points

1. In Figure S3C, PRIMPOL knockdown further exacerbates the replication fork restart defect. Does RFWD3 knockdown produce a similar phenotype? Furthermore, does simultaneous inactivation of both RFWD3 and PRIMPOL lead to an additive or epistatic effect? Addressing these questions would provide further insights into the functional relationship between RFWD3 and PRIMPOL in replication fork restart.

Reviewer #3

(Remarks to the Author)

This manuscript explores the role of SLFN11 in modulating DNA replication stress responses. The authors propose that SLFN11 expression interferes with the RFWD3-PRIMPOL-mediated DNA damage tolerance pathway, thereby inhibiting efficient replication fork restart after stalling events. Through a combination of single-molecule DNA fiber assays and super-resolution microscopy, they report that RFWD3 facilitates PRIMPOL recruitment to stalled forks, a process that is antagonized by SLFN11.

The overall concept of the RFWD3-PRIMPOL axis as a key player in DNA damage tolerance is interesting and could provide valuable insights into the mechanisms of replication stress response, with potential implications for therapeutic strategies targeting replication stress in cancer. However, the current dataset hardly provides any mechanistic details. Additional experiments and more comprehensive data analysis are necessary to solidify the proposed mechanistic links between SLFN11 and RFWD3-PRIMPOL and to convincingly demonstrate the role of SLFN11 in this pathway.

Major Comments

1. The manuscript claims that PRIMPOL is essential for replication fork restart via the RFWD3-PRIMPOL axis. However, this contradicts reported findings:

At 2 mM HU, Agata Smogorzewska et al. (Nature Communications, 2024) demonstrated that fork restart occurs independently of PRIMPOL.

2. Recent studies have elucidated the critical roles of various SLFN11 domains—such as its RNase activity, ATPase motifs, ssDNA binding domains, and phosphorylation sites—in mediating its cellular functions. The authors should investigate how these domains contribute to SLFN11-mediated suppression of RFWD3 recruitment/ activity or its role in replication fork restart. If the authors' conclusions are correct, any of the RNase domain, the ssDNA-binding domain, ATPase activity, and phosphorylation sites—all essential for DNA damage responses—may provide an essential link in the context of RFWD3 regulation. Boon, N. J., et al., *Science*, 384, 785–792 (2024); Zhang, P., et al., *Sci. Immunol.*, 9, eadj5465 (2024); Ogawa A., et al., *Mol. Cell*, (2025).

3. The authors provided insufficient evidence linking PRIMPOL recruitment to RPA Ubiquitination. The authors seem to suggest that PRIMPOL recruitment to stalled replication forks is mediated by RFWD3 and may involve RPA ubiquitination. In Figure 1C, they show that SLFN11 suppresses RPA32 ubiquitination under replication stress, implying a connection between SLFN11 and RFWD3 function. However, the manuscript does not conclusively demonstrate that the observed band represents true ubiquitination of RPA32. Ubiquitin-specific pull-down assays (for example, using His-Ub constructs) should be conducted to confirm SLFN11 really affects RPA ubiquitination. Furthermore, since multiple residues on RPA are known to be ubiquitinated by RFWD3, and RPA70 and PCNA are also a ubiquitination target, it is interesting to test whether the observed changes are specific to RPA32 or part of a broader modification pattern. The authors should clarify how these modifications are influenced by SLFN11 (and its mutant) and RFWD3.

4. The manuscript seems to suggest that PRIMPOL recruitment is mediated by RFWD3 via RPA ubiquitination. The author should provide mechanistic data that show how RFWD3 can facilitate PRIMPOL recruitment (via RPA ubiquitination?).

5. The data is primarily derived from HAP1 cells. The authors should confirm whether the observed effects of SLFN11 on RFWD3 function are reproducible in other cell lines, particularly RPE cells.

6. The authors present data in Figure 4A suggesting that SLFN11 expression reduces RFWD3 recruitment to replication forks using super-resolution microscopy. However, it is well-documented that RPA foci increase in SLFN11 knockout(KO) cells, theoretically enhancing the platform for RFWD3 recruitment. Given this, it remains unconvincing that RFWD3 recruitment decreases in the presence of SLFN11. They should provide more data. For example, to determine if SLFN11 disrupts their interaction, the authors should perform RPA-RFWD3 co-immunoprecipitation (Co-IP) and/or proximity ligation assays (PLA) to assess interactions such as those between RPA-RFWD3 and the nascent DNA strands (e.g, incorporated IdU)-RFWD3. Identifying which SLFN11 domains affect these interactions would clarify the mechanism.

7. The manuscript claims that RFWD3 knockdown does not affect replication fork restart in HAP1 cells, contradicting Inano et al. (2017, *Molecular Cell*), who demonstrated that RFWD3 knockout in HAP1 cells leads to strong phenotypes, including defects in homologous recombination (HR) repair. The authors fail to reconcile the more severe phenotypes seen in RFWD3 KO models. Moreover, Inano et al. showed that RFWD3's ubiquitination activity is not required for HU tolerance, which may be inconsistent with the proposed RFWD3-PRIMPOL axis (if the authors think the RPA ubiquitination is the key event). The authors may want to investigate whether ubiquitination-deficient RFWD3 mutants affect PRIMPOL localization and replication fork recovery under HU treatment.

8. The manuscript does not consider the role of RFWD3 downstream factors, such as MCM8/9 and HROB, which are critical for replication fork recovery and homologous recombination.

9. Other than lowered RFWD3 recruitment, the manuscript does not explore potential mechanisms underlying the reduced RFWD3-mediated ubiquitination. Is it possible that SLFN11 directly impairs RFWD3's ubiquitin ligase activity or that deubiquitinating enzymes (DUBs) are recruited or activated in the presence of SLFN11? Another plausible explanation is a reduction in RFWD3 expression levels, as suggested by Western blot data in Figure 1F and Figure 4A. The authors should clarify whether SLFN11 affects RFWD3 at the transcriptional level or through protein degradation by evaluating RFWD3 mRNA levels and protein stability.

10. Mu Y. et al. (2016) demonstrated that SLFN11 deficiency enhances overall homologous recombination (HR) repair efficiency, suggesting that RFWD3 is just one of several HR-related proteins influenced by SLFN11. The manuscript focuses narrowly on RFWD3 without considering SLFN11's broader regulatory effects on other HR factors, such as RAD51 or BRCA1/2.

11. The manuscript shows that SLFN11 knockout leads to increased gaps in nascent DNA, a phenotype that is PRIMPOL-dependent. Additionally, RFWD3 and PRIMPOL knockdowns in SLFN11 KO cells produce similar phenotypes (Figure 1G and Figure 2C), supporting the proposed SLFN11-RFWD3-PRIMPOL axis. However, to conclusively define their relationship, double knockdown (DKO) experiments are needed. For instance, analyzing whether SLFN11/RFWD3 DKO cells exhibit additive or overlapping phenotypes upon PRIMPOL knockdown would clarify whether these proteins act in the same pathway. Data from Figure S2E could be expanded to include PRIMPOL depletion in SLFN11/RFWD3 DKO cells.

12. The manuscript lacks sensitivity assays to confirm the functional consequences of SLFN11-dependent fork instability. If SLFN11 suppresses RFWD3-PRIMPOL, HU sensitivity assays in SLFN11 KO and SLFN11/PRIMPOL double KO cells are essential to determine whether PRIMPOL loss rescues SLFN11-dependent defects.

13. Figure 3A: Whether SLFN11 expression affects RFWD3 protein levels should be explicitly shown, as Figures 1F and 4A suggest potential changes in RFWD3 expression.

14. Figure 3B: The manuscript does not provide data on the distribution of SLFN11 and RPA in the same experimental system. Including this comparison would strengthen the conclusions regarding their interaction.

15. Figure 3E: γ H2AX Western blot data would be valuable to support the immunofluorescence findings. The figure order and corresponding text descriptions need to be clarified for consistency.

Minor Comments

1. Figure 3E: It appears that quantitative γ H2AX data were intended to be presented here, but this may have been shifted to Figure 3G.

2. Figure 4A: The expression of FLAG-RFWD3 decreases with DOX treatment, but this has not been normalized. The authors should normalize the total FLAG signal to the PCNA-proximal FLAG signal and present the data as a ratio.

3. Figure 4B: Western blot data for PRIMPOL expression are missing. Similar to RFWD3, the authors should normalize the total cellular PRIMPOL signal to the PCNA-proximal PRIMPOL signal and present it as a ratio.

4. Figure 4D: Cellular images are missing in this panel. Additionally, SLFN11 expression decreases drastically with siRFWD3, making it difficult to evaluate the results. Clarification is needed to ensure proper interpretation.

5. Figure 4E: The figure lacks mechanistic insights and fails to meet the standards for a Nature Communications publication.

6. Inconsistent Statement – Page 7, Line 6 from Bottom:

The manuscript refers to Figure S4E as HAP1 cell data in the text, but the figure legend indicates it represents RPE cell data. This inconsistency should be corrected.

Version 1:

Reviewer comments:

Reviewer #1

(Remarks to the Author)

In this manuscript, the authors investigate how SLFN11 modulates replication fork restart upon replication stress caused by stalled forks. Using a combination of DNA fiber assays, and Single-Molecule Localization Microscopy (SMLM), the authors demonstrate that SLFN11 binds RPA and limits RFWD3 recruitment and PRIMPOL-mediated gapped DNA synthesis, thereby restraining replication fork restart. Finally, they demonstrate that PRIMPOL mitigates replication-associated DNA damage in the absence of SLFN11, and that SLFN11's ATPase and RPA-binding activities are critical for its inhibitory function.

The manuscript provides insights into SLFN11-mediated regulation of replication fork restart in the RFWD3-PRIMPOL axis. The experiments are well-executed, and many of the previous review concerns have been addressed. While the manuscript demonstrates that SLFN11 restricts replication fork restart by reducing RFWD3 and PRIMPOL association with chromatin, mechanistic insights are limited. Specifically, the precise mechanism by which SLFN11 limits RFWD3/PRIMPOL recruitment remains unclear. The requirement of SLFN11's ATPase activity for downstream inhibition of RFWD3/PRIMPOL also remains unresolved. Additionally, the functional interplay between RFWD3 and PRIMPOL has not been fully dissected.

For the stated focus of the paper, how SLFN11 expression alters fork restart efficiency through regulation of the RFWD3-PRIMPOL DNA damage tolerance axis, the evidence provided supports the key conclusions. However, further mechanistic work could strengthen their model.

Some concerns are noted below:

The additional AP-MS and SMLM data strengthen the claim that SLFN11 indirectly influences RFWD3 activity via reduced chromatin association. However, the current evidence isn't sufficient to address how SLFN11 prevents RFWD3 recruitment. The "mislocalization" mechanism proposed does not clarify whether this is due to competition with RPA or due to changes in chromatin structure/accessibility. Given that RFWD3 loss affects multiple chromatin-bound proteins, it would be interesting to test whether SLFN11 selectively disrupts RFWD3-RPA interactions or globally impacts RFWD3's chromatin engagement. Co-IP for RPA-RFWD3 in SLFN11-expressing cells could also directly address this point.

The authors' conclusion that "PRIMPOL likely works downstream of RFWD3" is reasonable, but the mechanistic detail underlying this relationship remains unclear. It is not known whether RFWD3 promotes PRIMPOL recruitment directly via ubiquitinated RPA. Is it possible to generate a PRIMPOL mutant that is capable of localizing to chromatin independently of RPA. Can this mutant restore fork restart in RFWD3-depleted cells? This could help address whether RFWD3's role is limited to recruitment or does it affect PRIMPOL's catalytic activity.

For our original question: The authors must provide additional compelling evidence confirming that PRIMPOL's role in fork restart depends on RFWD3. For instance, do stalled forks persist longer when RFWD3 and PRIMPOL are depleted compared to RFWD3 knockdown alone?

The authors' explanation in the text improves clarity, but it still does not fully address the original concern about whether PRIMPOL's function in fork restart depends on RFWD3. The authors' argument is largely correlative (loss of RFWD3 reduces PRIMPOL recruitment), but functional dependence cannot be demonstrated without performing double knockdowns of RFWD3 and PRIMPOL.

Figure 3G: The authors state, "Next, we found that increased DNA damage, as marked by gH2AX signal, was associated with forks that cannot fully restart due to either pathway suppression by SLFN11 or when PRIMPOL was knocked down in cells lacking SLFN11 expression".

Details of how this quantification was done are unclear. Does the average gH2AX density take into account EdU-labeled DNA across the entire nucleus? Or is it quantified per individual fork? How can one distinguish whether the observed DNA damage arises specifically from forks that fail to restart versus from replication-independent lesions? Would quantifying gH2AX intensity at individual EdU foci help clarify this relationship? Similarly, the details for Figure 3F should be clearly stated.

This reviewer remains unconvinced that there is no difference in the BrdU/PI profile. The Y-axis in S1b shows lower excitation in SLFN11 KO cells compared to the control.

Further, based on S2b, the SLFNKO are not true KO and are hypomorphic lines. This is important because the interpretation can alter based on the levels of the SLFN11 in the cells.

Figure 4a: Are there increased levels of PRIMPOL in SLFN11 KD? Also, the siRNA is very inefficient.

Dox induction to induce levels of SLFN11 in cells that lack this is not physiological. This experiment has many caveats, and it is recommended that the authors discuss this.

Why is PCNA looking rod-like in Figure 3b compared to the -dox control?

For 4e: Can the authors show γ -H2AX in a control experiment?

Results section title and text describing Figure 4: Persistent SLFN11 localization to RPA inhibits replication fork restart "To determine whether SLFN11 directly antagonizes replication fork restart, we analyzed by SMLM the ability of SLFN11 to dynamically bind replication forks in untreated, HU, and restart conditions."

It is not fully clear how the SMLM density measurements of SNFL11 at RPA foci quantitatively correlate with functional inhibition of fork restart without a functional assay like DNA fiber analysis.

Discussion: The authors mention "it is unclear whether SLFN11 directly competes with RFWD3 at forks" The new SMLM data show that SLFN11-WT co-localizes with RPA in an HU-dependent manner and that this interaction is ATPase-dependent, correlating with inhibition of RFWD3 localization and fork restart. This provides a mechanistic link supporting the model that SLFN11 may compete with RFWD3-PRIMPOL for RPA-coated ssDNA.

However, the precise competitive mechanism remains unresolved. It is still unclear whether SLFN11 directly displaces RFWD3/PRIMPOL from RPA, or if it sterically prevents binding. Additionally, the regulation of this competition, for instance, whether post-translational modifications of SLFN11 or RPA influence binding affinity, is not addressed. While the authors note this as an open question, it would strengthen their claim if at least preliminary evidence or discussion of potential regulatory mechanisms were provided.

Finally, it would be useful to address whether the ATPase activity of SLFN11 is required solely for RPA binding or also for downstream inhibition of RFWD3/PRIMPOL recruitment, as this distinction impacts the interpretation of the competition model. Overall, the experiments support the proposed model, but further mechanistic dissection is needed to fully elucidate the competition hypothesis.

Reviewer #2

(Remarks to the Author)

While the authors have partially addressed my previous concerns, several points remain speculative. In particular, the mechanistic details of how SLFN11 disrupts the RFWD3–PRIMPOL axis to impair replication fork recovery are not fully elucidated. Nonetheless, considering the overall depth and quality of the data presented, I support the publication of this manuscript.

Reviewer #3

(Remarks to the Author)

The authors have largely improved the manuscript in this revision, but I still have some concerns regarding the mechanistic interpretation. My concern centers on the role of SLFN11's ssDNA-binding interface at stalled forks. The authors report that an ssDNA-binding–site mutant (K652D) still blocks fork restart, implying that ssDNA binding is dispensable. However, previous publications (Mu 2016, Murai group, Lammens group, and so on) emphasized that SLFN11's binding to RPA-coated ssDNA at stalled forks is fundamental for its fork-blocking function. In those studies, the ssDNA-binding mutant abolished SLFN11's ability to associate with RPA/ssDNA and to sensitize cells to replication stress. If K652D here truly shows no loss of function, that contradicts with those findings.

To strengthen their data, it is desirable to provide direct evidence regarding K652D's localization and function at stalled forks. For example, can the authors show the SMLM analysis (as in Fig. 4A) for SLFN11 K652D at RPA foci during HU treatment and recovery? If ssDNA binding is dispensable, one would expect K652D to accumulate at RPA-marked stalled forks just as well as wild-type SLFN11, whereas an ATPase-dead mutant would not. If SLFN11's fork-blocking activity is truly independent of direct ssDNA binding, the manuscript should explore what alternative mechanism or interaction allows SLFN11 to act at stalled forks. Does SLFN11 bind RPA through a specific motif (or domain) distinct from the DNA-binding site? Could SLFN11 be competing with RFWD3 for an RPA-binding surface, or does it require its ATPase domain to remodel something at the fork? A Co-IP experiment in nuclease-treated extracts to test whether SLFN11 K652D still pulls down RPA would support a protein–protein interaction model.

In Fig. 4A, RPA foci/density decreased after HU→wash. If this reflects SLFN11-dependent competition/exclusion at the RPA platform, then the K652D mutant should phenocopy WT. It would be interesting to show the HU→wash fold-change in RPA for no-dox, WT, K652D, and ATPase-dead with identical normalization.

Version 2:

Reviewer comments:

Reviewer #3

(Remarks to the Author)

The authors have addressed all the raised questions.

Re: Nature Communications manuscript NCOMMS-25-04807

Here is our point-by-point response to the Reviewers' Comments (Author's comments/rebuttal in blue):

REVIEWER COMMENTS

Reviewer #1 (Remarks to the Author):

Replication stress is caused by the slowing or stalling of replication forks and can happen for many reasons, including DNA damage, replication-transcription conflicts, fork defects, dNTP imbalance etc. A stalled fork can be restarted by various mechanisms, including fork reversal, DNA repair pathways, and recruitment of factors that can stabilize and help re-initiation of DNA replication. In the current manuscript, the authors study SLFN11, a factor with tRNA endoribonuclease and DNA binding functions, and its role in replication fork dynamics, especially in response to replication stress. Using DNA fiber analysis, the authors propose that SLFN11 antagonizes a fork recovery pathway involving the E3 ligase, RFWD3, and the human DNA primase-polymerase (PRIMPOL). The data is clean and, for the most part, convincing. However, mechanistic details on how SLFN11 counteracts the RFWD3-PRIMPOL axis were missing. The entire manuscript is based on interesting observations from DNA fiber analysis, but none of the experiments get to the mechanism of action. The observations made in this manuscript are interesting but too preliminary.

We appreciate the Reviewer's comments stating that the data is clean, and for the most part, convincing. We recognize that our initial study did not fully explore the mechanism of action of how SLFN11 counteracts the RFWD3-PRIMPOL axis for fork restart. In our revised manuscript, we provide new experiments to address how SLFN11 regulates RFWD3-PRIMPOL localization and function using SLFN11 structure/function analysis (see responses below).

Some specific comments are noted below:

1) Figure S1B: There is a noticeable difference in the EdU vs DAPI flow data between the control and SLFN11 KO. There is a population of cells between 2C and 4C content (with no excitation on the Y-axis), suggesting that replication is progressing at a very slow rate or cells are in the S-phase, but the time given for EdU incorporation is insufficient. Further, late S phase seems to be impacted. It is incorrect to state that there are no detectable changes.

We apologize for the confusion in this initial FACS analysis. We have repeated this study several times and the data showed no apparent cell cycle defect when comparing parental HAP1 cells to *SLFN11* KO HAP1 cells, using FACS analysis (see new Supplementary Fig. 1b).

2) For all the DNA fiber experiments, a thorough quantification of the IdU-only tracts must be presented. The length of CldU after HU would strengthen their argument about whether the replication upon restart was impacted or not.

We have calculated IdU tracts for all of the DNA fiber analysis that was done throughout the manuscript (the numbers are available in the Resource Data file). IdU tracts (no HU treatment) are generally not affected by the loss of SLFN11. This was demonstrated in Fig. 1a when different SLFN11 KO lines were analyzed for replication fork speed (length of CldU tract in untreated cells). As you may be aware, DNA fibers that only contain IdU tracts (IdU-only tracts that don't have CldU signal) signify % of stalled replication forks (with little or no fork restart). We showed that in SLFN11-expressing cells, there are more stalled forks (IdU-only tracts) that

accumulate after recovery from HU treatment (Supplementary Fig. 2c). Similarly, in our DNA fiber analysis, we calculated the length of CldU tracts after HU treatment (wash off) to show the efficiency of fork restart as a ratio of CldU/IdU tract lengths throughout our study. Both the CldU and IdU tract length numbers are available in the Resource Data file. For greater clarity, we graphed the CldU tract lengths for *SLFN11* KO cells (as requested by the Reviewer), and have now put the data in Supplementary Fig. 2a. The trend for CldU tract length differences are generally reflected in the CldU/IdU tract length ratios.

3) Figure 1C: The finding that ATR signaling remains intact, but RPA32 ubiquitination is altered in *SLFN11*-expressing cells is intriguing. What is the potential mechanism linking *SLFN11* to RFWD3-mediated ubiquitination? Does *SLFN11* directly or indirectly influence RFWD3 activity? Is there any evidence for an interaction between *SLFN11* and RFWD3 (based on IPs or Alpha-fold predictions)?

To address this question, we performed AP-MS of FLAG-RFWD3 and showed that it can be found associated with different DNA-binding, chromatin regulators, and DNA repair proteins upon HU treatment, but only in the absence of *SLFN11* (see new Fig. 5c and Supplementary Fig. 7a). We did not detect *SLFN11* as one of the co-precipitating proteins of FLAG-RFWD3 in parental cells (*SLFN11*-expressing). This suggests that RFWD3 and *SLFN11* do not themselves stably interact, rather it is likely that *SLFN11* inhibits the ability of RFWD3 to associate with chromatin-bound proteins, including RPA. This partially explains why there is less RPA ubiquitination when *SLFN11* is expressed in cells; RFWD3 is no longer localized to the chromatin/replication fork in the presence of *SLFN11*. We also used SMLM imaging technique as an orthogonal approach to show that ectopic expression of *SLFN11* in cells that don't normally express *SLFN11* can also lead to reduced chromatin association of FLAG-RFWD3 at the forks (new Fig. 5a-b). Based on these set of new data, we propose that *SLFN11* likely indirectly influences RFWD3 activity by mis-localizing or preventing RFWD3 from associating with chromatin, thus, reducing the ubiquitination levels of RPA.

4) Figure 1D: the authors need to provide better quantifiable data. Is there a dose-dependent change to RPA Ub when *SLFN11* is induced?

The purpose of this experiment is to confirm that Ub-RPA32 is diminished when *SLFN11* expression is reconstituted in *SLFN11* KO cells. This rules out potential artifacts related to CRISPR-Cas9 gene disruption or single cell cloning that would be independent of *SLFN11* protein expression. We also provide new experiments to observe *SLFN11*-mediated inhibition of Ub-RPA32 in HEK293T cells using IP-Western of Flag-Ub expression in a HU-dependent manner (new Fig. 1f).

5) Figure 1F: Which band represents RFWD3? Is there an upregulation of RFWD3 in cells lacking *SLFN11*, and is this increase resulting in enhanced ubiquitination of RPA32?

The specific and non-specific bands are now clearly annotated in the immunoblots, we apologize for this oversight. We don't generally observe an increase or changes in RFWD3 protein levels in cells lacking *SLFN11* (see Supplementary Fig. 5a).

6) Figure 1G: The authors demonstrate that RFWD3 is required for efficient fork restart in *SLFN11* KO cells. However, it remains unclear whether RFWD3 also affects fork elongation rates post-restart. As mentioned in #2, quantification of fiber data is critical. Measuring elongation in RFWD3-depleted cells will help determine if its role is specific to restart or if its depletion causes a broader replication defect. If RFWD3 depletion also slows elongation, the

observed effects may reflect a general impairment in replication progression rather than a restart-specific function.

RFWD3 has a well-established role in promoting fork restart (Elia et al, *Mol Cell*, 2015). We have now calculated the IdU tracts for RFWD3 siRNA knockdown cells (untreated) and compared it to ctrl siRNA (untreated) in HAP1 parental and *SLFN11* KO cells (see new Supplementary Fig. 2f). The data shows that IdU tract lengths are all relatively similar regardless of RFWD3 and/or *SLFN11* loss. This shows that RFWD3 depletion does not cause general fork speed slowdown; thus, we don't believe that RFWD3 depletion also affects fork elongation rates post-restart, as we do not observe a general impairment in replication progression.

7) The reviewer is not convinced with the following statement: Together, these results confirm that the replication fork restart defect in *SLFN11*-expressing cells is consistent with insufficient RFWD3 activity. Why is the red tract starting and then not continuing? Is it taking longer to restart, or does it only progress to a certain extent once it restarts? For example, 1G- how many never fire after HU? And does that change upon the loss of *SLFN11*? And what happens when RFWD3 is co-depleted?

To avoid confusion and mis-statements, we modified the text to, "Together, these results show that RFWD3 is required for replication fork restart in cells lacking *SLFN11* expression." (see page 6 last paragraph). Related to "why red tract is starting and then not continuing, or whether it progresses to a certain extent once it restarts", these are interesting questions that cannot be addressed with current tools. To my knowledge, tracking individual fork movement to measure tract length in real-time in live cells is currently impossible to do based on currently available tools. Based on our new data, we propose that *SLFN11* likely blocks accessibility of RPA-coated ssDNA to RFWD3 and PRIMPOL, and thus prevents efficient fork restart. *SLFN11* could be rate-limiting in cells, and therefore, *SLFN11* may be competing with RFWD3 for the RPA substrate. The presence of shorter red tracts (CldU) could be due to incomplete inhibition by *SLFN11*. Thus, in our DNA fiber analysis, we would expect a Gaussian distribution of shorter sizes of red tracts. And this is what we observed in Fig. 1h and Supplementary Fig. 2g. In the future, it will be interesting to test whether *SLFN11* can kick off the replisome of restarted forks by limiting or competing with PRIMPOL engagement. In this study, what we refer to as restarted forks are not a result of a newly fired replication origins. TLS polymerases can bypass lesions to mitigate fork stalling to enable fork restart from the same replisome/fork. This does not require new origin firing (dormant origins) from an adjacent site. We are only tracking forks that were previously labeled by IdU, and then determining whether those previously elongating forks can restart (CldU) after HU treatment for DNA fiber analysis.

8) Another statement that needs clarification: Taken together, these data indicate that ssDNA gaps in *SLFN11* knockout cells are likely dependent on PRIMPOL engagement downstream of RFWD3 function to restart replication forks. What is the proof that it is downstream of RFWD3? The defect could also be caused by a global defect in replication restart caused by RFWD3 depletion.

We showed that in the absence of *SLFN11*, depletion of RFWD3 reduced the engagement of PRIMPOL at replication forks by STORM imaging (see Fig. 5g). This suggests that RFWD3 is required for the proper localization of PRIMPOL during fork restart. It is likely that RFWD3 has other global effects on replication fork and DNA repair dynamics (see new data Fig. 5c). How RFWD3 regulates chromatin binding of other proteins and DNA repair is beyond the scope of

this study. As it relates to replication fork restart, PRIMPOL engagement is likely downstream of RFWD3 function to restart replication forks. Since at this point of the study, both RFWD3 and PRIMPOL loss of function studies phenocopy each other (Fig 1-2), but epistasis has not been established yet until Fig. 5. Thus, we will tone down this statement.

9) Figure S4C: Can a positive control be included?

This western blot panel has been repeated with a positive control (DU145) for SLFN11 expression (new Supplementary Fig. 4c).

10) Figure 3C-D: Are these images only representing early S-phase cells? How is the quantification in C and D done? What is EdU density? Or does this represent the number of EdU foci?

For these panels, only S-phase nuclei were scored, indicated by abundant, homogeneous, and nuclear PCNA staining. This has been added to the main text for clarity.

SMLM quantification from our group used here builds on established methods, assays and computational tools that are described in great detail in our previous publications and well as work from other labs (see PMID: 21926998, PMID: 22384026, and particularly PMID: 35365626, see Methods **Correlation functions for analysis of SMLM data**, and **Supplementary Note 1, PMID: 34473946, Supplemental information, Methods S3-S4, and also PMID: 30631072 SI note 2, and Supp Fig 2. PMID: 25843623**) and all the technical and experimental parameters are now well-established and have been thoroughly evaluated by expert reviewers and meticulously validated in prior works. Specifically, previous work from our group utilized SMLM and downstream auto-correlation analysis to measure EdU density per nucleus, as well as the average density of labeled targets of interest (PMID: 35365626).

The EdU density calculated in PCNA+ nuclei present the coordinates of individual fluorescent EdU molecules that were localized with an accuracy of several nanometers (shown as single pixels). SMLM data substantially differs from the data generated via conventional epi-fluorescence or confocal microscopy approaches that measure diffraction-limited intensity distribution over multiple pixels, where measurements of co-localization or overlap between two objects/foci/clusters typically relies on simple segmentation of clusters boundaries.

In the manuscript we utilize a robust and unbiased data-mining statistical approach that measures the pair-wise distances between every molecule of EdU to all other EdU molecules, which is computed directly from the molecular coordinates. Based on all the measured pair-wise distances the algorithm then automatically generates the probability distribution of the distances (probably density) using a cross-correlation (or pair-correlation) function. To simplify, this provides a robust statistical measure for the distance distribution that only converge if there is non-random probability whereby a number of molecules are within a certain range (or distribution) of distances from other molecules. Importantly, if molecules are placed at random distances the probability will be zero even at incredibly high-density of molecules since it will yield similar frequency for all distances, whereas even a small subset of molecules (within a field of high-density of randomly localized molecules) are positions within a range of distances from one another will yield a non-zero probability. This means that if there are few non-random events (statistically significant) where pairs of molecules are within (or near) a distance of 20 nm or 30 nm of each other, these are considered as being associated with the same process or complexes, even though their specific molecular coordinates might not overlap as these are

resolved at 10 nm. In our measurements of molecular complexes at replication forks and repair complexes, the range of distances we obtained are less than 100 nm, which also reflect the spread of the SMLM signals including that of EdU at individual forks. Importantly, these distances are far below the diffraction limit of light, and as such will be considered as co-localization of overlapping clusters when analyzed using standard confocal or epi-fluorescence microscopy.

11) Figure 3E: What does the image provided represent? Is it a control cell or upon dox induction?

The representative image provided is a control nucleus (no dox, RPE). SMLM images present the coordinates of individual fluorescent molecules that were localized with an accuracy of several nanometers (shown as single pixels).

12) How is the extent of induction of SLFN11 controlled? Does the extent of induction impact primpol function?

The extent of induction is controlled by the concentration of dox in media and the time under treatment. We chose these parameters to achieve a near physiological level of SLFN11 expression (see, e.g. Fig. 1d).

13) Figure 3G: The reviewer is unconvinced about the image quality of gH2AX. Is this a single section? What is the resolution of SMLM? Why is PCNA not showing complete overlap with EdU, considering that PCNA can localize to replication and repair sites?

All Samples were illuminated in HILO mode which generates a quasi-lightsheet that excites a thin slice (300nm) in the sample plane thereby resulting in enhances out-of-plane rejection ensuring that any observed signals are obtained from the same plane. Further out-of-focus signal is rejected through the SMLM localization algorithm (**PMID: 34473946**). In-depth description of the technical parameters of the methods, experimental assays and analysis are provided in **PMID: 34473946, Supplemental information, Methods S3-S4 as well as in our other work (for example in PMID: 33953191, PMID: 33370257, PMID: 32542039, PMID: 31570834, PMID: 30631072 , PMID: 30422114, PMID: 30250272)**.

Statistically significant STORM events are where pairs of molecules are within (or near) a distance of 20 nm or 30 nm of each other. In our measurements of molecular complexes at replication forks and repair complexes, the range of distances we obtained are less than 100 nm. SMLM quantification from our group used here builds on established methods, assays and computational tools that are described in great detail in our previous publications and well as work from other labs (see PMID: 21926998, PMID: 22384026, and particularly PMID: 35365626, see Methods **Correlation functions for analysis of SMLM data**, and **Supplementary Note 1, PMID: 34473946, Supplemental information, Methods S3-S4, and also PMID: 30631072 SI note 2, and Supp Fig 2. PMID: 25843623**) and all the technical and experimental parameters are now well-established and have been thoroughly evaluated by expert reviewers and meticulously validated in prior works. Specifically, previous work from our group utilized SMLM and downstream auto-correlation analysis to measure EdU density per nucleus, as well as the average density of labeled targets of interest (PMID: 35365626).

The representative SMLM shown is the amalgamation of all stochastic SMLM fluorescent events are recorded over 2000 x 30-ms frames (2 minutes). SMLM images present the coordinates of individual fluorescent molecules that were localized with an accuracy of several nanometers (shown as single pixels at a given intensity). From a technical stance, EdU that does not localize to PCNA in the representative image could be underrepresented EdU molecules that are hindered by more abundant PCNA molecules in the representative image, but are taken into account in analysis (see above). From a biological stance, the kinetics of PCNA chromatin dynamics specifically in the context of replication restart are not clear, and there are other mechanisms contributing to chromatin-bound PCNA that could be relevant in the context of restart (i.e. post-replicative repair) that are beyond the scope of these experiments.

14) The data presented in the figure doesn't match what is mentioned in the text: SMLM analysis of PRIMPOL reveals that PRIMPOL recruitment to forks is compromised by knockdown of RFWD3, which is consistent with our findings that knockdown of PRIMPOL did not impact RFWD3-mediated RPA ubiquitination (Figure 4D).

We have revised the text to offer more clarity in this statement. Fig. 5g shows that RFWD3 siRNA knockdown (but not control siRNA), in the absence of SLFN11, reduced the number of PRIMPOL molecules or density per focus during fork restart conditions. Importantly, SLFN11 expression prevented the recruitment of PRIMPOL to forks irrespective of RFWD3 levels. This suggests that RFWD3 is required for PRIMPOL recruitment at replication forks to mediate fork restart only in cells that are SLFN11-deficient. In Fig. 2b, PRIMPOL siRNA knockdown has no effect on RPA ubiquitination. Since RPA ubiquitination is mediated by RFWD3 function at the fork, this experiment shows that PRIMPOL likely works downstream of RFWD3.

15) The authors must provide additional compelling evidence confirming that PRIMPOL's role in fork restart depends on RFWD3. For instance, do stalled forks persist longer when RFWD3 and PRIMPOL are depleted compared to RFWD3 knockdown alone?

Our best case for PRIMPOL's role in fork restart and its dependency on RFWD3 is our use of orthogonal functional assays (DNA fiber analysis and SMILM) to show similar effects on fork restart defects with either PRIMPOL or RFWD3 loss-of-function studies. Additionally, RFWD3 siRNA knockdown prevents efficient recruitment of PRIMPOL to chromatin/replication forks (see Fig. 5g). This suggests that RFWD3 functions upstream of PRIMPOL in the same pathway. To truly delve into epistasis, we would need to generate single, double and triple CRISPR-induced KO cells for SLFN11, RFWD3, and PRIMPOL, to properly assess each of their contributions in cell sensitivity/growth, cell cycle, and replication dynamics assays. These experiments are currently in the pipeline but due to time and funding constraints, we were not able to finish these experiments in a timely manner. The focus of the paper is more on how SLFN11 expression alters fork restart efficiency through the regulation of the RFWD3-PRIMPOL DNA damage tolerance axis. We need to generate more tools to study how each of these factors operate in totality for replication stress response.

16) The proposed model is oversimplified and leaves unanswered questions regarding the precise mechanisms by which SLFN11 influences RFWD3 and PRIMPOL recruitment and function during replication fork restart. For instance, does SLFN11 directly inhibit RFWD3's E3 ligase activity, or how does it impact the localization of RFWD3 and PRIMPOL to the fork?

Our new data supports that SLFN11 expression decreases RFWD3 and PRIMPOL association/localization to the fork, and that RFWD3 depletion can also decrease PRIMPOL association to the fork. This model is consistent with the RFWD3 I639K mutation observation in

a Fanconi anemia patient that disrupts RPA32 winged helix domain binding, recruitment to stalled forks, and substrate ubiquitination without direct loss of catalytic activity. We predict that it is unlikely that SLFN11 can directly inhibit RFWD3's E3 ligase activity. Future biochemical and structural studies on RFWD3 and SLFN11 interaction on stalled forks may provide deeper insights into this important question. Our updated model now reflects this new data (new Fig. 6a).

17) Minor comment: Within the introduction, the description of SLFN11's proposed functions based on previous literature is quite dense and lists multiple mechanisms in rapid succession. This section could be broken into chunks, and specifying which mechanisms are more relevant would provide clarity.

The most relevant mechanism is the model that proposes an irreversible fork restart defect proposed by Murai and colleagues. Our manuscript explains this apparent defect by demonstrating that while forks can restart in the presence of SLFN11, they are significantly shorter owing to a lack of gapped synthesis by PRIMPOL. The introduction has been revamped according to the Reviewer's suggestions.

Also, RFWD3 and PRIMPOL have not been sufficiently introduced. Since they are central to the study, briefly describing their known roles in replication fork restart before discussing their connection to SLFN11 would improve context.

Our approach to the Introduction is to explain what was known at the start of our studies and to provide rationale for the line of inquiry. Since RFWD3 and PRIMPOL were not previously known to be connected to SLFN11 biology, we felt it was smoother to introduce them throughout the Results section. Otherwise, the introduction of unrelated fork-binding factors would feel a bit contrived. This is only our opinion, but would be happy to change the Introduction according to the Reviewer's suggestions if needed.

Reviewer #2 (Remarks to the Author):

In this manuscript, the authors demonstrate that SLFN11 expression impairs replication fork restart by inhibiting the RFWD3-PRIMPOL DNA damage tolerance axis, a pathway critical for resolving stalled replication forks, promoting replication stress recovery, and maintaining genomic stability. They further show that SLFN11 suppresses RFWD3-PRIMPOL localization at restarted forks and that PRIMPOL recruitment to replication forks is largely dependent on RFWD3.

While these findings are of potential interest to the field, the manuscript lacks mechanistic insights into how SLFN11 disrupts the RFWD3-PRIMPOL axis to inhibit replication fork recovery. Without a more detailed mechanistic characterization, the study remains preliminary. Therefore, I consider the manuscript insufficiently developed for publication in its current form.

Major Points

1. The proposed counteraction of the RFWD3-PRIMPOL axis by SLFN11 at stalled replication forks is intriguing; however, the precise molecular mechanism remains unclear. Since SLFN11, RFWD3, and PRIMPOL are all RPA-interacting proteins and are recruited to stalled forks via RPA, could SLFN11 compete with RFWD3 and PRIMPOL for RPA binding, thereby preventing their localization to stalled forks? If so, how is this competition regulated? Further investigation into the interplay between SLFN11, RFWD3, and PRIMPOL at RPA-coated forks is necessary to substantiate this model.

We appreciate the comments by the Reviewer. We now provide new experiments to address whether SLFN11-WT or the ATPase (helicase) mutant interacts or co-localizes with RPA using SMLM analysis (new Fig. 4 and Supplementary Fig. 6). We now show that SLFN11-WT interacts (co-localizes) with RPA in an HU-dependent manner (new Fig. 4a-c). Interestingly, upon fork restart, the interaction of SLFN11-WT with RPA is further elevated. This nicely correlates with the chromatin/fork localization of SLFN11 in response to replication stress. In contrast, the ATPase mutant does not associate with RPA in response to HU treatment, nor does it persist during fork restart (Supplementary Fig. 6a-b). Importantly, we show that the ATPase mutant is incapable of inhibiting fork restart and RFWD3 localization at replication forks (Fig. 4g and Supplementary Fig. 6c), suggesting that the ATPase domain of SLFN11 is critical in counteracting RFWD3-PRIMPOL axis at stalled forks via its RPA interaction. These new experiments provide additional details and mechanistic insights into how SLFN11 regulates fork restart. Based on our findings, we believe that future studies may now begin to biochemically tease apart how SLFN11 is able to compete away RFWD3-PRIMPOL binding to RPA-coated ssDNA using *in vitro* reconstitution assays. Additionally, SLFN11 could be regulated by post-translational modifications (PTMs); this is currently an open question in the field. Nevertheless, the localization of SLFN11 to stalled replication forks is clearly dynamic.

2. Although the authors demonstrate that RFWD3 is required for PRIMPOL recruitment to stalled forks, the underlying mechanism is not addressed. Does RFWD3 directly mediate PRIMPOL recruitment through physical interaction, post-translational modifications, or modulation of RPA dynamics? Additional experiments are needed to delineate this regulatory pathway.

Our study is the first to show that RFWD3 is required for PRIMPOL recruitment to stalled forks in response to HU restart (new Fig. 5g). Whether this is mediated through direct binding of PRIMPOL to RFWD3 or through indirect means via PTMs is a difficult question to address without some fishing experiments. We initially tried to determine whether dynamic RFWD3 phosphorylation can be captured using phospho-proteomics analysis. While we were able to get complete coverage of candidate phosphorylation sites by digesting with a variety of proteases, we did not identify any HU induced phosphorylation at either of the putative SQ motifs proposed regulate RFWD3 activity. Our current hypothesis is that RFWD3 likely regulates the RPA dynamics through its ubiquitination status. Potentially, ubiquitinated RPA enhances its turnover on ssDNA to enable PRIMPOL to reprime and initiate DNA synthesis for fork restart. We hope to address this question in greater detail in future studies.

3. In Figures S2D-E, the authors show that RFWD3 knockout (KO) clones derived from SLFN11 KO cell lines exhibit a phenotype similar to the HAP1 parental SLFN11-expressing cells. To determine whether RFWD3's E3 ligase activity is critical for this effect, the authors should perform complementation assays by reintroducing wild-type RFWD3 and an E3 ligase-inactive mutant into RFWD3 KO cells. This would clarify whether the observed phenotype is dependent on RFWD3-mediated ubiquitination.

We would be surprised if the role of RFWD3 in fork restart is independent of its ubiquitin ligase activity. Nevertheless, we attempted this experiment, as requested by the Reviewer, but we encountered some technical difficulties trying to complement siRNA-resistant forms of RFWD3 WT and C315A (cat mutant) back into RFWD3 siRNA knockdown RPE1 cells. We thought this would be a more worthwhile experiment to attempt since we can then use SMLM analysis to assess fork restart in complemented cells, amongst other co-localization and protein interaction analysis in an intact cell. We did not find a commercially available RFWD3 antibody that worked for immunofluorescence. Although we were unsuccessful in generating complemented cell lines,

we were finally able to ectopically express FLAG-WT RFWD3 and FLAG-(C315A mutant) RFWD3 in RPE1 cells to assess how SLFN11 expression can alter RFWD3's localization at replication forks by SMLM analysis (Fig. 5b). Both FLAG-WT RFWD3 and FLAG-(C315A) RFWD3 localization or enrichment at forks were compromised by SLFN11 expression. Importantly, an ATPase mutant form of SLFN11 could not disrupt the localization of FLAG-RFWD3 (Supplementary Fig. 6c). We also provided new data showing that the RFWD3 interactome is greatly affected by SLFN11 expression. For this experiment, we used the FLAG-(C315A mutant) RFWD3 to assess catalytic activity independent binding of RFWD3 to chromatin-binding proteins by AP-MS technique (Fig. 5c).

Minor Points

1. In Figure S3C, PRIMPOL knockdown further exacerbates the replication fork restart defect. Does RFWD3 knockdown produce a similar phenotype? Furthermore, does simultaneous inactivation of both RFWD3 and PRIMPOL lead to an additive or epistatic effect? Addressing these questions would provide further insights into the functional relationship between RFWD3 and PRIMPOL in replication fork restart.

Knockdown of either PRIMPOL or RFWD3 in SLFN11-deficient cells produces a similar fork restart defect, as measured by CldU tract lengths immediately after HU wash off. The difference for CldU tract length is quite dramatic. While there is an increase in the frequency of stalled forks (previously elongating forks that don't restart at all after HU wash off) in either SLFN11-expressing parental cells, or in SLFN11-deficient cells with PRIMPOL knockdown, the difference isn't as dramatic. We anticipate this would be the same for RFWD3 knockdown, but did not check this because this phenotype was relatively mild in comparison to the tract length phenotype. We decided to focus more on tract length changes during fork restart. Both PRIMPOL knockdown (Fig. 2c) and RFWD3 knockdown (Fig. 1h) have similar effects on dramatically reducing the tract lengths of restarted forks in SLFN11-deficient cells at comparable levels to SLFN11-expressing HAP1 cells (Fig. 1h). It would be somewhat difficult to assess epistasis relationship if we compare the effects of individual versus combined knockdown of both PRIMPOL and RFWD3, and then try to determine whether there is an additive effect or not, especially when the tract lengths are already quite short. This point by the Reviewer is well-taken, and in future studies we hope to understand whether RFWD3 and PRIMPOL are indeed epistatic by looking at other phenotypic effects, including cell survival or cell cycle effects, rather than fork restart. For this study, instead of determining a genetic relationship between RFWD3 and PRIMPOL, we wanted to assess whether RFWD3 could be functioning upstream of PRIMPOL as it relates to their localization at replication forks. In parallel, we wanted to address whether SLFN11 expression could alter the localization or interaction of RFWD3 and PRIMPOL at replication forks. We spent most of our energy trying to understand this question by setting up SMLM experiments to assess the dependency of their localization at replication forks (Figs. 3-5).

Reviewer #3 (Remarks to the Author):

This manuscript explores the role of SLFN11 in modulating DNA replication stress responses. The authors propose that SLFN11 expression interferes with the RFWD3-PRIMPOL-mediated DNA damage tolerance pathway, thereby inhibiting efficient replication fork restart after stalling events. Through a combination of single-molecule DNA fiber assays and super-resolution microscopy, they report that RFWD3 facilitates PRIMPOL recruitment to stalled forks, a process that is antagonized by SLFN11.

The overall concept of the RFWD3-PRIMPOL axis as a key player in DNA damage tolerance is interesting and could provide valuable insights into the mechanisms of replication stress response, with potential implications for therapeutic strategies targeting replication stress in

cancer. However, the current dataset hardly provides any mechanistic details. Additional experiments and more comprehensive data analysis are necessary to solidify the proposed mechanistic links between SLFN11 and RFWD3-PRIMPOL and to convincingly demonstrate the role of SLFN11 in this pathway.

Major Comments

1. The manuscript claims that PRIMPOL is essential for replication fork restart via the RFWD3-PRIMPOL axis. However, this contradicts reported findings: At 2 mM HU, Agata Smogorzewska et al. (Nature Communications, 2024) demonstrated that fork restart occurs independently of PRIMPOL.

We do not believe there is a contradiction in the results coming from two distinct studies, as the conditions that are used for these studies were different, leading to different results. The experiment I think the Reviewer is referring to in the Smogorzewska and colleagues study (Conti et al, Nat Comm, 2024) was done at 4 mM HU and the restarted forks were assessed after knockdown of fork-binding proteins (DDI1/2, RTF2), and the restarted forks were assessed after 90 min of DNA synthesis (CldU). These conditions are quite different from our study where we are observing dramatic fork restart tract length only after 30 min. Importantly, the concentration of HU utilized in the experiment is also critical for assessing DNA synthesis levels, as higher levels of HU causes oxidative stress and acute depletion of ribonucleotide levels with the higher HU dose may be more difficult for full recovery after HU wash off. For our study, we used 2 mM HU, whereas in the Conti et al study, 4 mM HU was used. It is also possible that distinct regions of the genome may have different fork restart regulation (PRIMPOL-dependent and -independent functions). For replisomes blocked by RTF2 or DDI1/2 binding, these factors may be present in regions of the genome that are inaccessible to the RFWD3-PRIMPOL machinery, depending on how the protein-DNA barriers are processed for fork recovery.

2. Recent studies have elucidated the critical roles of various SLFN11 domains—such as its RNase activity, ATPase motifs, ssDNA binding domains, and phosphorylation sites—in mediating its cellular functions. The authors should investigate how these domains contribute to SLFN11-mediated suppression of RFWD3 recruitment/ activity or its role in replication fork restart. If the authors conclusions are correct, any of the RNase domain, the ssDNA-binding domain, ATPase activity, and phosphorylation sites—all essential for DNA damage responses—may provide an essential link in the context of RFWD3 regulation. Boon, N. J., et al., *Science*, 384, 785–792 (2024); Zhang, P., et al., *Sci. Immunol.*, 9, eadj5465 (2024); Ogawa A., et al., *Mol. Cell*, (2025).

Thank you for this helpful suggestion. We have now included new data in the revised manuscript that investigates several of these separation-of-function SLFN11 mutations. The results from these experiments helped us to conclude that while ssDNA binding and tRNA hydrolysis SLFN11 activities are dispensable for inhibiting PRIMPOL-mediated fork restart, the ATPase domain is required for SLFN11 accumulation at RPA-containing replication forks and for inhibiting fork restart (Fig. 4 and Supplementary Fig. 6a-b). Additionally, mutation of the ATPase domain compromises the ability of SLFN11 to inhibit RFWD3 localization to replication forks (Supplementary Fig. 6c).

3. The authors provided insufficient evidence linking PRIMPOL recruitment to RPA Ubiquitination. The authors seem to suggest that PRIMPOL recruitment to stalled replication forks is mediated by RFWD3 and may involve RPA ubiquitination. In Figure 1C, they show that SLFN11 suppresses RPA32 ubiquitination under replication stress, implying a connection between SLFN11 and RFWD3 function. However, the manuscript does not conclusively

demonstrate that the observed band represents true ubiquitination of RPA32. Ubiquitin-specific pull-down assays (for example, using His-Ub constructs) should be conducted to confirm SLFN11 really affects RPA ubiquitination. Furthermore, since multiple residues on RPA are known to be ubiquitinated by RFWD3, and RPA70 and PCNA are also a ubiquitination target, it is interesting to test whether the observed changes are specific to RPA32 or part of a broader modification pattern. The authors should clarify how these modifications are influenced by SLFN11 (and its mutant) and RFWD3.

We appreciate this experimental suggestion by the Reviewer. We performed FLAG-Ub pulldown assays to confirm that SLFN11 expression does indeed reduce ubiquitinated RPA32 (Fig. 1f). We do not observe any changes to ubiquitinated PCNA (data not shown). It would not be surprising for RFWD3 to target additional substrates at the replication fork. We also performed an AP/MS experiment using the catalytic mutant of RFWD3 as bait to show that SLFN11 expression suppresses interaction of RFWD3 to other chromatin-binding factors (Fig. 5c). It is possible that some of these interacting protein complexes may also be targeted for ubiquitination by RFWD3. In future studies, we plan to delve into how RFWD3-mediated ubiquitination of replication fork-binding factors can modulate DNA repair or replication stress response in cells.

4. The manuscript seems to suggest that PRIMPOL recruitment is mediated by RFWD3 via RPA ubiquitination. The author should provide mechanistic data that show how RFWD3 can facilitate PRIMPOL recruitment (via RPA ubiquitination?).

PRIMPOL does not encode a ubiquitin binding domain, so it is unlikely that recruitment is directly mediated by Ub-RPA32. We think it is likely indirect; it is possible that increased Ub-RPA turnover at stalled replication forks could enhance PRIMPOL initiation by limiting the levels of RPA on ssDNA, as excess RPA has been shown to inhibit PRIMPOL priming using *in vitro* assays (Guilliam et al, *NAR* 2015; Martinez-Jimenez et al, *Sci Rep*, 2017).

5. The data is primarily derived from HAP1 cells. The authors should confirm whether the observed effects of SLFN11 on RFWD3 function are reproducible in other cell lines, particularly RPE cells.

The data on replication fork restart using DNA fiber analysis are reproducible in multiple cell lines, including RPE cells. We then used single-molecule imaging (SMLM) to capture fork restart (EdU density at forks) changes in RPE cells in the presence or absence of SLFN11 (Fig. 3a-b), and showed the effects of SLFN11 on RFWD3 localization at replication forks (Fig. 5b).

6. The authors present data in Figure 4A suggesting that SLFN11 expression reduces RFWD3 recruitment to replication forks using super-resolution microscopy. However, it is well-documented that RPA foci increase in SLFN11 knockout(KO) cells, theoretically enhancing the platform for RFWD3 recruitment. Given this, it remains unconvincing that RFWD3 recruitment decreases in the presence of SLFN11. They should provide more data. For example, to determine if SLFN11 disrupts their interaction, the authors should perform RPA-RFWD3 co-immunoprecipitation (Co-IP) and/or proximity ligation assays (PLA) to assess interactions such as those between RPA-RFWD3 and the nascent DNA strands (e.g, incorporated IdU)-RFWD3. Identifying which SLFN11 domains affect these interactions would clarify the mechanism.

Using SMLM analysis, we now show that SLFN11 localization or enrichment near RPA at replication forks increases upon HU treatment, and this is further enhanced after HU wash off, indicative of an increase in SLFN11 binding to RPA-containing forks during fork restart

conditions (Fig. 5b-c). The increased binding of SLFN11 to RPA-containing forks will likely inhibit RFWD3 interaction with RPA, thus preventing the ubiquitination and turnover of RPA. We also have some preliminary evidence that in the absence of SLFN11, RFWD3 depletion causes an increase in RPA density at forks (Figure 1A, below), while knockdown of PRIMPOL has no effect on RPA density (Figure 1B, below). This is likely because RFWD3 works upstream of RPA ubiquitination and stabilization, while PRIMPOL is downstream. The RFWD3 cat mutant also leads to more RPA density at foci, in comparison to RFWD3 WT (Figure 1C, below). However, in the presence of SLFN11, RFWD3 cat mutant cannot promote more RPA density at foci (Figure 1C, below). Since this was not done in a RFWD3 knockout/knockdown-complementation system, the effect of the RFWD3 cat mutant is likely due to a dominant-negative effect in the partial suppression of RPA ubiquitination and protein turnover. Clearly much work is needed to understand the interplay and mechanism behind the role of RFWD3 in mediating RPA turnover in SLFN11-deficient cells. We feel this is too preliminary to add as data to this current study.

Figure 1. (A) RPE cells were treated with control siRNA or siRFWD3 without dox treatment. HU-induced fork stalling, PBS wash, and release into fresh media with EdU was performed prior to SMLM of RPA. Scatterplot quantification measuring the average RPA density per focus in PCNA+ nuclei in siCt and siRFWD3 RPEs. *p* values of technical replicates calculated using unpaired two-tailed t-test from two biological replicates (siCt: N=110, siRFWD3: N=145). Error bars = mean, SEM. (B) RPE cells were treated with control siRNA or siPRIMPOL without dox treatment. HU-induced fork stalling, PBS wash, and release into fresh media with EdU was performed prior to SMLM of RPA. Scatterplot quantification measuring the average RPA density per focus in PCNA+ nuclei in siCt and siPRIMPOL RPEs. *p* values of technical replicates calculated using unpaired two-tailed t-test from two biological replicates (siCt: N=122, siPRIMPOL: N=68). Error bars = mean, SEM. (C) Untreated or DOX-treated RPEs were transiently transfected with WT- or Cat mutant-RFWD3-FLAG for 24h. Scatterplot quantification measuring pair-correlation analysis of the average RPA density in PCNA+ nuclei for each condition. *p* values of technical replicates calculated using unpaired two-tailed t-test from two biological replicates (No Dox, WT: N=127; No Dox, mutant: N=135; Dox, WT: N=100; Dox, mutant: N=98). Error bars = mean, SEM.

We also performed separation-of-function SLFN11 expression experiments as described in response to point #2 above.

7. The manuscript claims that RFWD3 knockdown does not affect replication fork restart in HAP1 cells, contradicting Inano et al. (2017, Molecular Cell), who demonstrated that RFWD3 knockout in HAP1 cells leads to strong phenotypes, including defects in homologous recombination (HR) repair. The authors fail to reconcile the more severe phenotypes seen in RFWD3 KO models. Moreover, Inano et al. showed that RFWD3's ubiquitination activity is not required for HU tolerance, which may be inconsistent with the proposed RFWD3-PRIMPOL axis (if the authors think the RPA ubiquitination is the key event). The authors may want to investigate whether ubiquitination-deficient RFWD3 mutants affect PRIMPOL localization and replication fork recovery under HU treatment.

The Reviewer is missing the point that in the study by Inano et al, they use HAP1 cells for exploring the role of RFW3 in DNA repair. In these cells, SLFN11 is still expressed, thus this blunts and sensitizes the cells to DNA damaging agents and other fork perturbations. Clearly RFW3 have multiple roles as it is a Fanconi Anemia protein that modulates DNA repair and replication stress response in cells. In our study, we are trying to tease apart the mechanism of how modulating SLFN11 expression causes a rewiring of the DNA damage tolerance axis, leading to differences in how cells recover from stalled replication forks. The RFW3 function we are trying to characterize in this study is one that is operational when SLFN11 is absent in cells. The role of RFW3 in DNA repair in SLFN11-positive cells is not the point of this study. Several points: 1) the referenced paper doesn't explore fork restart at all and does not even contain the word 'restart'. 2) We show that RFW3 depletion has very significant effects on fork restart, but only if SLFN11 is absent in cells, consistent with our model in which RFW3 is functionally repressed by SLFN11; 3) The results with the C315A mutant are very difficult to interpret because all of the data points are greater than the EC90.

8. The manuscript does not consider the role of RFW3 downstream factors, such as MCM8/9 and HROB, which are critical for replication fork recovery and homologous recombination.

The Reviewer's point is taken. In future studies we will consider whether MCM8/9 and HROB acts downstream of RFW3 in the context of fork restart in SLFN11-deficient cells.

9. Other than lowered RFW3 recruitment, the manuscript does not explore potential mechanisms underlying the reduced RFW3-mediated ubiquitination. Is it possible that SLFN11 directly impairs RFW3's ubiquitin ligase activity or that deubiquitinating enzymes (DUBs) are recruited or activated in the presence of SLFN11? Another plausible explanation is a reduction in RFW3 expression levels, as suggested by Western blot data in Figure 1F and Figure 4A. The authors should clarify whether SLFN11 affects RFW3 at the transcriptional level or through protein degradation by evaluating RFW3 mRNA levels and protein stability.

We took a very close look at how SLFN11 might affect RFW3 protein steady state levels. Under dox induction or in knockout cells, SLFN11 expression had no meaningful impact on steady state levels of RFW3 so there is no strong rationale to evaluate mRNA or protein stability (new Supplementary Fig. 5a).

10. Mu Y. et al. (2016) demonstrated that SLFN11 deficiency enhances overall homologous recombination (HR) repair efficiency, suggesting that RFW3 is just one of several HR-related proteins influenced by SLFN11. The manuscript focuses narrowly on RFW3 without considering SLFN11's broader regulatory effects on other HR factors, such as RAD51 or BRCA1/2.

Our study is focused on investigating the fork restart defect caused by SLFN11 and not on DNA repair. The enhancement of overall HR repair efficiency in SLFN11-deficient cells is still not well-understood.

11. The manuscript shows that SLFN11 knockout leads to increased gaps in nascent DNA, a phenotype that is PRIMPOL-dependent. Additionally, RFW3 and PRIMPOL knockdowns in SLFN11 KO cells produce similar phenotypes (Figure 1G and Figure 2C), supporting the proposed SLFN11-RFW3-PRIMPOL axis. However, to conclusively define their relationship, double knockdown (DKO) experiments are needed. For instance, analyzing whether SLFN11/RFW3 DKO cells exhibit additive or overlapping phenotypes upon PRIMPOL

knockdown would clarify whether these proteins act in the same pathway. Data from Figure S2E could be expanded to include PRIMPOL depletion in SLFN11/RFWD3 DKO cells.

Since siRFWD3 knockdown or RFWD3 KO in SLFN11 KO cells already showed near complete loss of fork restart (as measured by tract lengths), it may be quite challenging to interpret the results of additional depletion of PRIMPOL on top of RFWD3 and SLFN11 loss. We don't think the phenotype for the reduced DNA synthesis during fork restart with either PRIMPOL or RFWD3 loss in SLFN11-deficient cells could be further exacerbated with additional loss of factors. However, we understand that it would be informative to determine epistasis for the role of RFWD3 and PRIMPOL with respect to fork restart. While each of the factors may contribute in other ways to limit genomic instability, we predict that RFWD3 and PRIMPOL are indeed part of the same DNA damage tolerance axis. Another way to address this question mechanistically, we tested whether RFWD3 is required for PRIMPOL recruitment/enrichment at restarted forks. In Fig. 5g, we showed that RFWD3 knockdown reduces PRIMPOL density per focus in PCNA (+) nuclei. Additionally, the increase in PRIMPOL recruitment occurs only when SLFN11 is absent; if SLFN11 is expressed, the recruitment/enrichment of PRIMPOL at restarted forks becomes suppressed, and is no longer dependent on RFWD3. This suggests that RFWD3 functions upstream of PRIMPOL, but only in the absence of SLFN11.

12. The manuscript lacks sensitivity assays to confirm the functional consequences of SLFN11-dependent fork instability. If SLFN11 suppresses RFWD3-PRIMPOL, HU sensitivity assays in SLFN11 KO and SLFN11/PRIMPOL double KO cells are essential to determine whether PRIMPOL loss rescues SLFN11-dependent defects.

We appreciate this experimental suggestion by the Reviewer. The proposed dose-response experiments can now be found in new Fig. 2d. These show that either RFWD3 or PRIMPOL KO in *SLFN11* KO cells can partially reverse HU resistance in *SLFN11* KO cells.

13. Figure 3A: Whether SLFN11 expression affects RFWD3 protein levels should be explicitly shown, as Figures 1F and 4A suggest potential changes in RFWD3 expression.

Please refer to the response to critique #9.

14. Figure 3B: The manuscript does not provide data on the distribution of SLFN11 and RPA in the same experimental system. Including this comparison would strengthen the conclusions regarding their interaction.

Thank you for this comment. This data can now be found in new Fig. 4c, where we analyze SLFN11 recruitment/enrichment at replication forks near RPA signals.

15. Figure 3E: γ H2AX Western blot data would be valuable to support the immunofluorescence findings. The figure order and corresponding text descriptions need to be clarified for consistency.

Generally, we find that Western blot data for γ H2AX levels is not as quantitative as SMLM analysis, as there could be multiple bands of H2AX that may be associated with H2AX ubiquitination. Some of the ubiquitinated H2AX forms are also phosphorylated at the Ser139 site. Also, we are focused more on DNA damage immediately adjacent or in the vicinity of replication forks undergoing fork restart, The advantage of SMLM is that we can directly look at these replication fork-associated DNA damage sites, as opposed to the assessment of overall γ H2AX levels on chromatin.

Minor Comments

1. Figure 3E: It appears that quantitative γ H2AX data were intended to be presented here, but this may have been shifted to Figure 3G.

We apologize for this lack of clarity, and have edited the text accordingly.

2. Figure 4A: The expression of FLAG-RFWD3 decreases with DOX treatment, but this has not been normalized. The authors should normalize the total FLAG signal to the PCNA-proximal FLAG signal and present the data as a ratio.

We have redone this experiment, and now it shows that FLAG-RFWD3 WT and cat mutant expression do not decrease with Dox treatment (new Fig. 5a).

3. Figure 4B: Western blot data for PRIMPOL expression are missing. Similar to RFWD3, the authors should normalize the total cellular PRIMPOL signal to the PCNA-proximal PRIMPOL signal and present it as a ratio.

Western blot data for PRIMPOL expression (with or without Dox treatment) can be found in Figs. 3e, Supplementary Fig. 3d, and Supplementary Fig. 4a-c. We are confused by what the Reviewer is asking for here. There are no dox-induced effects on PRIMPOL expression in multiple cell lines tested.

4. Figure 4D: Cellular images are missing in this panel. Additionally, SLFN11 expression decreases drastically with siRFWD3, making it difficult to evaluate the results. Clarification is needed to ensure proper interpretation.

We have edited the text to acknowledge the decrease in SLFN11 expression with siRFWD3 knockdown. We don't think this changes the interpretation of the data since this experiment is addressing whether PRIMPOL can be captured by SMLM analysis to be enriched at restarted forks in the absence of SLFN11, and whether RFWD3 knockdown can alter the recruitment/enrichment of PRIMPOL at these sites. In the presence of SLFN11 (dox treatment), PRIMPOL recruitment becomes suppressed, regardless of whether RFWD3 is depleted or not.

5. Figure 4E: The figure lacks mechanistic insights and fails to meet the standards for a Nature Communications publication.

We apologize for the lack of mechanistic insight in the model. We have revised this model to more clearly depict the RFWD3 and PRIMPOL recruitment defect in the context of SLFN11-expressing cells (new Fig. 6a).

6. Inconsistent Statement – Page 7, Line 6 from Bottom:

The manuscript refers to Figure S4E as HAP1 cell data in the text, but the figure legend indicates it represents RPE cell data. This inconsistency should be corrected.

Thank you for pointing this out - it has been corrected in the revised manuscript.

manuscript NCOMMS-25-04807A (please see author's response in blue)

2nd Rebuttal letter

Reviewer #1:

In this manuscript, the authors investigate how SLFN11 modulates replication fork restart upon replication stress caused by stalled forks. Using a combination of DNA fiber assays, and Single-Molecule Localization Microscopy (SMLM), the authors demonstrate that SLFN11 binds RPA and limits RFWD3 recruitment and PRIMPOL-mediated gapped DNA synthesis, thereby restraining replication fork restart. Finally, they demonstrate that PRIMPOL mitigates replication-associated DNA damage in the absence of SLFN11, and that SLFN11's ATPase and RPA-binding activities are critical for its inhibitory function.

The manuscript provides insights into SLFN11-mediated regulation of replication fork restart in the RFWD3-PRIMPOL axis. The experiments are well-executed, and many of the previous review concerns have been addressed. While the manuscript demonstrates that SLFN11 restricts replication fork restart by reducing RFWD3 and PRIMPOL association with chromatin, mechanistic insights are limited. Specifically, the precise mechanism by which SLFN11 limits RFWD3/PRIMPOL recruitment remains unclear. The requirement of SLFN11's ATPase activity for downstream inhibition of RFWD3/PRIMPOL also remains unresolved. Additionally, the functional interplay between RFWD3 and PRIMPOL has not been fully dissected.

For the stated focus of the paper, how SLFN11 expression alters fork restart efficiency through regulation of the RFWD3-PRIMPOL DNA damage tolerance axis, the evidence provided supports the key conclusions. However, further mechanistic work could strengthen their model.

We are pleased that Reviewer #1 stated that the experiments are well-executed, and many of the previous review concerns have been addressed, and that we have provided enough evidence to support the key conclusions in our revised manuscript. This suggests that our revised paper, as seen by three reviewers, have addressed all the points necessary to support our key conclusions and that the revised manuscript should now be suitable for publication at this time. Reviewer #1 noted that more mechanistic work could be done to expand on our key conclusions, which is the case for any mechanism. The wish list of experiments raised by Reviewer #1 often references experiments that already appear in the revised manuscript. Those that extend beyond the stated scope of the paper are not limitations of the key conclusions of our present work; rather, they are natural and exciting directions for follow-up research, which we entirely agree with. However, we would argue that expansion of the scope at this late stage would unnecessarily delay publication of a paper that is well-supported by the existing experimental evidence.

Some concerns are noted below:

The additional AP-MS and SMLM data strengthen the claim that SLFN11 indirectly influences RFWD3 activity via reduced chromatin association. However, the current evidence isn't sufficient to address how SLFN11 prevents RFWD3 recruitment. The "mislocalization" mechanism proposed does not clarify whether this is due to competition with RPA or due to changes in chromatin structure/accessibility. Given that RFWD3 loss affects multiple chromatin-bound proteins, it would be interesting to test whether SLFN11 selectively disrupts RFWD3-RPA interactions or globally impacts RFWD3's chromatin engagement. Co-IP for RPA-RFWD3 in SLFN11-expressing cells could also directly address this point.

The short answer to the Reviewer's question is that SLFN11 does both, it likely disrupts RFW3-RPA interaction as shown by our STORM analysis that RFW3 localization at the fork is inhibited upon SLFN11 expression, and that SLFN11 globally disrupts the localization and function of RFW3 through our AP-MS data (see Figure 5). In our manuscript, we already showed that upon SLFN11 expression, SLFN11 selectively localizes to RPA upon HU treatment, and this is further elevated during fork restart conditions (HU washout) (Fig 4C). To study this phenomenon with greater depth and detail, we use sophisticated and unbiased computational measurements of single-molecule protein localization to determine whether hard-to-detect or rare events are significantly different in our experimental conditions through the use of super-resolution imaging techniques in an intact cell. Our data demonstrates that SLFN11 is enriched with RPA at a higher level under conditions of fork restart. Under this same condition, we tested whether FLAG-RFW3 WT or E3 ligase defective mutant localization at forks can be inhibited by SLFN11 expression (Fig 5B). Based on our data, both WT and E3 ligase defective mutant localization at forks are severely compromised when SLFN11 is expressed. Again, we are using super-resolution microscopy to assess the probability of co-localization/co-enrichment in an unbiased manner that evaluates hundreds of data points independently. This suggests that RFW3-RPA interactions are disrupted at chromatin/forks, since RPA bound to ssDNA is primarily localized to stalled forks. We changed the language to make it less demonstrative, but our conclusions are consistent with an extensive body of literature on RFW3-RPA and SLFN11-RPA interactions.

The authors' conclusion that "PRIMPOL likely works downstream of RFW3" is reasonable, but the mechanistic detail underlying this relationship remains unclear. It is not known whether RFW3 promotes PRIMPOL recruitment directly via ubiquitinated RPA. Is it possible to generate a PRIMPOL mutant that is capable of localizing to chromatin independently of RPA. Can this mutant restore fork restart in RFW3-depleted cells? This could help address whether RFW3's role is limited to recruitment or does it affect PRIMPOL's catalytic activity.

For our original question: The authors must provide additional compelling evidence confirming that PRIMPOL's role in fork restart depends on RFW3. For instance, do stalled forks persist longer when RFW3 and PRIMPOL are depleted compared to RFW3 knockdown alone?

These are all interesting mechanistic questions that are beyond the scope of our current study. We plan to work on the role of ubiquitinated RPA and potentially other PTMs and how they impact PRIMPOL recruitment and function. These are questions related to general PRIMPOL function and fits outside of our current study. As for the proposed experiment by the Reviewer, this would not be definitive because of potential compensatory pathways that could obscure linear epistasis and could also be confounded by partial knockdown, as the reviewer points out below.

The authors' explanation in the text improves clarity, but it still does not fully address the original concern about whether PRIMPOL's function in fork restart depends on RFW3. The authors' argument is largely correlative (loss of RFW3 reduces PRIMPOL recruitment), but functional dependence cannot be demonstrated without performing double knockdowns of RFW3 and PRIMPOL.

We already provided evidence that PRIMPOL likely functions downstream of RFW3 in SLFN11-low cells: RFW3 is required for PRIMPOL to localize to forks (Fig. 5G), and DNA fiber assays give similar phenotypic results as RFW3 knockdown or PRIMPOL knockdown give the same phenotype in SLFN11-low cells. The reduced tract lengths (fork speed) during fork restart

is already very low in RFWD3 or PRIMPOL knockdown, so it is difficult to use this methodology for epistasis analysis. We have tweaked the text to soften the conclusion that PRIMPOL's function in fork restart is "largely dependent on RFWD3" as it regulates its localization to forks.

Figure 3G: The authors state, "Next, we found that increased DNA damage, as marked by gH2AX signal, was associated with forks that cannot fully restart due to either pathway suppression by SLFN11 or when PRIMPOL was knocked down in cells lacking SLFN11 expression".

Details of how this quantification was done are unclear. Does the average gH2AX density take into account EdU-labeled DNA across the entire nucleus? Or is it quantified per individual fork? How can one distinguish whether the observed DNA damage arises specifically from forks that fail to restart versus from replication-independent lesions? Would quantifying gH2AX intensity at individual EdU foci help clarify this relationship? Similarly, the details for Figure 3F should be clearly stated.

We have clarified the details of how quantification was done regarding the measure of the gH2AX density in the vicinity of EdU-labeled DNA (foci) across the entire nucleus of multiple cells. The gH2AX density at EdU sites accounts for all EdU foci in the nucleus, these are all active forks that produce nascent DNA (via EdU incorporation). The density of these EdU foci will be shorter or less when SLFN11 is expressed, and longer or greater when SLFN11 is reduced. These are the forks that we are interrogating to ask whether there is a correlation with more gH2AX (DNA damage) at these forks when the EdU density is shorter or less. Our results are in line with our hypothesis that hindered fork restart will lead to higher probability of fork associated DNA damage (gH2AX). Finally, PRIMPOL knockdown in SLFN11-low cells will reduce fork restart to similar levels as SLFN11-expressing cells, and we provide evidence of more fork -associated DNA damage (gH2AX) (Fig 4G). We have clarified the details with text for Fig. 3F and 3G.

This reviewer remains unconvinced that there is no difference in the BrdU/PI profile. The Y-axis in S1b shows lower excitation in SLFN11 KO cells compared to the control.

There was no significant difference that can be observed between the different samples in our experimental conditions.

Further, based on S2b, the SLFNKO are not true KO and are hypomorphic lines. This is important because the interpretation can alter based on the levels of the SLFN11 in the cells.

We believe that the reviewer misread the text. As stated in the figure legend, the HAP1 cells are subjected to siRNA knockdown of SLFN11, and showed that using different oligo siRNA sequences give similar phenotype as the CRISPR-mediated KO clones of SLFN11 in HAP1 cells. The siRNA knockdown data complements the KO clones experiment to ensure that the results are consistent regarding whether SLFN11 levels was transiently or constitutively reduced in cells.

Figure 4a: Are there increased levels of PRIMPOL in SLFN11 KD? Also, the siRNA is very inefficient.

The reviewer is likely referring to S4a (not the main Figure 4a). We do not observe any increase in PRIMPOL protein level with SLFN11 knockdown, or at the minimum, the very slight differences are not significant between the different cell lines (SW1271 and DU145). siRNA

knockdown efficiency also varies between different cell lines, and therefore we wanted to ensure the results are consistent when applying a highly sensitive method, such as DNA fiber analysis. The siRNAs knockdown efficiencies could be more robust for certain cell lines, but the phenotype was consistent across different cell lines, suggesting that even partial knockdown of PRIMPOL was sufficient to observe fork restart defects.

Dox induction to induce levels of SLFN11 in cells that lack this is not physiological. This experiment has many caveats, and it is recommended that the authors discuss this.

Dox is used to engineer the expression of proteins to assess immediate and controllable effects of protein expression for functional analysis. SLFN11 is expressed at physiological levels in the dox-inducible rescue cells. We understand that using cell lines that normally don't express SLFN11 has its caveats. We have addressed the rationale behind this experiment in the text.

Why is PCNA looking rod-like in Figure 3b compared to the -dox control?

PCNA within replication hubs are a conglomerate of proteins that can be detected at the single-molecule level using super resolution imaging techniques and supported by several of our previous publications (see Rothenberg and colleagues). The images that we pick are completely arbitrary and are used simply to show an example of the computational image of where the proteins are enriched within the nucleus. We kindly ask the Reviewer to refer to the extensive publications on this SMLM technique by the Rothenberg lab.

For 4e: Can the authors show γ -H2AX in a control experiment?

We are unsure what the Reviewer is referring to here.

Results section title and text describing Figure 4: Persistent SLFN11 localization to RPA inhibits replication fork restart

"To determine whether SLFN11 directly antagonizes replication fork restart, we analyzed by SMLM the ability of SLFN11 to dynamically bind replication forks in untreated, HU, and restart conditions."

It is not fully clear how the SMLM density measurements of SNFL11 at RPA foci quantitatively correlate with functional inhibition of fork restart without a functional assay like DNA fiber analysis.

The functional assay, such as the DNA fiber analysis, were done in Figure 3a to validate the RPE cells and the rescue of RPE cells with Dox-inducible SLFN11.

Discussion: The authors mention "it is unclear whether SLFN11 directly competes with RFWD3 at forks"

The new SMLM data show that SLFN11-WT co-localizes with RPA in an HU-dependent manner and that this interaction is ATPase-dependent, correlating with inhibition of RFWD3 localization and fork restart. This provides a mechanistic link supporting the model that SLFN11 may compete with RFWD3-PRIMPOL for RPA-coated ssDNA.

However, the precise competitive mechanism remains unresolved. It is still unclear whether SLFN11 directly displaces RFWD3/PRIMPOL from RPA, or if it sterically prevents binding. Additionally, the regulation of this competition, for instance, whether post-translational modifications of SLFN11 or RPA influence binding affinity, is not addressed. While the authors

note this as an open question, it would strengthen their claim if at least preliminary evidence or discussion of potential regulatory mechanisms were provided.

The preliminary evidence is provided in Figures 4 and 5, demonstrating in an intact cell that SLFN11 localizes with RPA upon fork restart conditions and that SLFN11 expression displaces both RFWD3 and PRIMPOL from the replication fork in response to HU-mediated fork restart. We have clarified this point in the discussion section.

Finally, it would be useful to address whether the ATPase activity of SLFN11 is required solely for RPA binding or also for downstream inhibition of RFWD3/PRIMPOL recruitment, as this distinction impacts the interpretation of the competition model. Overall, the experiments support the proposed model, but further mechanistic dissection is needed to fully elucidate the competition hypothesis.

The Reviewer may have missed the data in the Supplementary info: Figure S6A shows that the SLFN11 ATPase mutant can still be recruited to replication foci in a HU-dependent manner. Interestingly, upon restart, the ATPase mutant cannot be fully retained at the forks. Figure S6B shows that the ATPase mutant cannot colocalize to RPA in response to HU, thus the ATPase mutant is likely unable to bind to RPA. This validates that the ATPase mutant of SLFN11 is required for RPA binding under the conditions of replication stress. Figure S6C shows that the ATPase mutant cannot inhibit the localization of Flag-RFWD3.

Reviewer #2:

I agree with Reviewer #1 that further clarification of how SLFN11 expression alters fork restart efficiency through regulation of the RFWD3–PRIMPOL DNA damage tolerance axis would strengthen the manuscript. That said, I also recognize that the authors have already performed a substantial amount of work, and the evidence provided sufficiently supports the key conclusions of the study. In my view, requiring the authors to fully resolve all mechanistic questions at this stage would be beyond the scope of the present study. The data as they stand meet the standards for publication in *Nature Communications*.

We appreciate the Reviewer's comments and recognition that the authors have already performed a substantial amount of work, and the evidence provided sufficiently supports the key conclusions of the study. The Reviewer also states that the data as stand meet the standards for publication in *Nature Comm*.

Reviewer #3:

The new biochemical and super-resolution experiments show convincingly that SLFN11—through its ATPase activity—displaces RFWD3 from RPA, reduces RPA32 ubiquitination, and blocks PRIMPOL-dependent fork restart. These findings are now validated in multiple cell lines. Notably, the K652D mutant—defective in canonical ssDNA binding—still blocks RFWD3/PRIMPOL recruitment, indicating that SLFN11 competes with RPA through a mechanism largely independent of its ssDNA-binding site.

The authors have satisfactorily addressed all the major concerns raised in the original review.

We thank the Reviewer for his/her appreciation of the K652D mutant data that showed a surprising result that disruption of ssDNA binding activity of SLFN11 did not significantly affect

its function in blocking fork restart, indicating that SLFN11's ability to inhibit RFD3/PRIMPOL recruitment mechanistically is largely independent of its ssDNA-binding activity. This was an important point that we tried to make in our study. Instead, the critical SLFN11 activity lies in its ATPase activity (helicase).

The authors have largely improved the manuscript in this revision, but I still have some concerns regarding the mechanistic interpretation. My concern centers on the role of SLFN11's ssDNA-binding interface at stalled forks. The authors report that an ssDNA-binding-site mutant (K652D) still blocks fork restart, implying that ssDNA binding is dispensable. However, previous publications (Mu 2016, Murai group, Lammens group, and so on) emphasized that SLFN11's binding to RPA-coated ssDNA at stalled forks is fundamental for its fork-blocking function. In those studies, the ssDNA-binding mutant abolished SLFN11's ability to associate with RPA/ssDNA and to sensitize cells to replication stress. If K652D here truly shows no loss of function, that contradicts with those findings.

We wish to clarify that the Mu et al. 2016 paper did not investigate ssDNA binding because this property of SLFN11 was not known until the 2024 *Nature Communications* paper by the Lammens group. In the Mu et al. paper, they showed that a C-terminal deletion from amino acids 741-901, and to a lesser extent 580-740, was unable to bind to recombinant RPA1 (Fig. 2). These experiments did not address ssDNA binding and are difficult to interpret in this context as the extreme C-terminal deletion also encompasses the regulatory S753 phosphorylation site and Y722, involved in dimer interface I while the internal deletion encompasses K591 of the same interface as well as K652 and the Walker A and B motifs. In the referenced papers from the Lammens group by Metzner et al. and Kugler et al., there are no studies of fork-blocking function and no functional studies to compare ours to. In the iScience paper by Fujiwara et al., from the Murai group, they show that K652D or K652E mutations are unable to sensitize cells to camptothecin (CPT). We do not challenge these results. In Figure 2, they showed that the K562 (the cell line) cells expressing a variety of SLFN11 mutants including K652D (the ssDNA binding mutant) exposed to 100 nM CPT for 4 h led to reduced EdU incorporation. However, it is not possible to differentiate between fork restart and cells entering S phase from G1 from the data as shown. It is important to note that fork collapse from CPT treatment is a fundamentally different experiment and biological scenario than transient replication stress caused by HU to measure fork restart. In the same figure, they treated the cells, extracted, and performed immunofluorescence for SLFN11. In this experiment, they observed less chromatin bound SLFN11 in the context of the K652D mutant, but signal remains present on chromatin – multicolor SMLM experiments in our manuscript can resolve recruitment of individual molecules where foci formation assays cannot.

To strengthen their data, it is desirable to provide direct evidence regarding K652D's localization and function at stalled forks. For example, can the authors show the SMLM analysis (as in Fig. 4A) for SLFN11 K652D at RPA foci during HU treatment and recovery?

If ssDNA binding is dispensable, one would expect K652D to accumulate at RPA-marked stalled forks just as well as wild-type SLFN11, whereas an ATPase-dead mutant would not. If SLFN11's fork-blocking activity is truly independent of direct ssDNA binding, the manuscript should explore what alternative mechanism or interaction allows SLFN11 to act at stalled forks. Does SLFN11 bind RPA through a specific motif (or domain) distinct from the DNA-binding site? Could SLFN11 be competing with RFD3 for an RPA-binding surface, or does it require its ATPase domain to remodel something at the fork?

A Co-IP experiment in nuclease-treated extracts to test whether SLFN11 K652D still pulls down RPA would support a protein-protein interaction model.

In Fig. 4A, RPA foci/density decreased after HU→wash. If this reflects SLFN11-dependent competition/exclusion at the RPA platform, then the K652D mutant should phenocopy WT. It would be interesting to show the HU→wash fold-change in RPA for no-dox, WT, K652D, and ATPase-dead with identical normalization.

The questions that the Reviewer is eluding to relates to an exciting new axis of experiments to clarify the importance of ssDNA binding versus RPA binding that includes single molecule microscopy, biochemistry, and structural biology. These are very interesting questions but are currently too difficult to resolve given the tools at hand. Specifically, the literature currently lacks a separation-of-function mutation in SLFN11 that abolishes RPA binding while preserving ssDNA binding, and there is no structural information on which to base a candidate approach. Importantly, our study is mainly focused on the SLFN11 ATPase mutant (helicase activity) as this mutant, in contrast to the K652D (ssDNA binding activity) or the tRNA hydrolysis mutant, demonstrated that it could not block fork restart as measured by SMLM analysis. It is conceivable that the K652D mutant lead to other replication stress or DDR defects (CPT sensitivity) that our studies using fork restart assays are not fully capturing. Future studies can help clarify different roles for each of SLFN11's different functional domains using a suite of replication stress and DDR assays. This was not the intention for our current study. In our study, we are focused on addressing fork restart mechanisms and how SLFN11 (through its ATPase activity) can antagonize downstream RFW3 and PRIMPOL-mediated gap filling function to help alleviate replication-associated DNA damage events. The detailed and specific experiments the Reviewer is proposing here is for our study to also try to reconcile and validate the negative control we used for our experiment (K652D mutant) when assessing the function of SLFN11 for fork restart mechanisms. Our study does not refute or support an older model proposed for SLFN11's ssDNA binding and chromatin function in a previously published study (Murai et al, 2018). This is not the intent of our current study and the experiments asked by the Reviewer is beyond the scope of this revised study, as we have already addressed the original critiques from the Reviewers.